# Polypyrimidine tract-binding proteins are essential for B cell development

**Elisa Monzón-Casanova[1,2]\*, Louise S Matheson[1], Kristina Tabbada[3], Kathi Zarnack[4], Christopher WJ Smith[2], Martin Turner[1]\***

[1]Laboratory of Lymphocyte Signalling and Development, The Babraham Institute, Cambridge, United Kingdom; [2]Department of Biochemistry, University of Cambridge, Cambridge, United Kingdom; [3]Next Generation Sequencing Facility, The Babraham Institute, Cambridge, United Kingdom; [4]Buchmann Institute for Molecular Life Sciences, Goethe University Frankfurt, Frankfurt am Main, Germany

**Abstract** Polypyrimidine tract-binding protein 1 (PTBP1) is a RNA-binding protein (RBP) expressed throughout B cell development. Deletion of *Ptbp1* in mouse pro-B cells results in upregulation of PTBP2 and normal B cell development. We show that PTBP2 compensates for PTBP1 in B cell ontogeny as deletion of both *Ptbp1* and *Ptbp2* results in a complete block at the pro-B cell stage and a lack of mature B cells. In pro-B cells PTBP1 ensures precise synchronisation of the activity of cyclin dependent kinases at distinct stages of the cell cycle, suppresses S-phase entry and promotes progression into mitosis. PTBP1 controls mRNA abundance and alternative splicing of important cell cycle regulators including CYCLIN-D2, c-MYC, p107 and CDC25B. Our results reveal a previously unrecognised mechanism mediated by a RBP that is essential for B cell ontogeny and integrates transcriptional and post-translational determinants of progression through the cell cycle.

**\*For correspondence:**
elisa.monzon-casanova@
babraham.ac.uk (EMó-C);
martin.turner@babraham.ac.uk
(MT)

**Competing interests:** The authors declare that no competing interests exist.

## Introduction

Antibody diversity is crucial to fight infections and is generated throughout B cell development by the orderly recombination of V(D)J gene segments of the immunoglobulin heavy (Igh) and light (Igl) chain loci. Igh-recombination at the pro-B cell stage typically occurs before Igl recombination in B cell ontogeny (*Figure 1—figure supplement 1A*). Pro-B cells can be separated into fractions (Fr) B and C according to Hardy's criteria (*Hardy et al., 1991*), with D to J gene segment recombination occurring in FrB pro-B cells prior to V to DJ recombination in FrC pro-B cells (*Hardy and Hayakawa, 2003*). Successful VDJ recombination of the Igh chain locus is coupled with a rapid proliferative expansion of early-pre-B cells (FrC') and the subsequent return to quiescence permissive for recombination at the Igl chain loci in late-pre-B cells (FrD) (*Herzog et al., 2009*). V(D)J recombination occurs during the G0/G1 phase of the cell cycle (*Schlissel et al., 1993*) and the RAG proteins, which are essential for V(D)J recombination, are degraded upon entry into S-phase, suppressing further recombination (*Li et al., 1996*). The alternation between proliferative and non-proliferative stages during B cell development is precisely controlled to maintain genomic integrity (*Herzog et al., 2009*; *Clark et al., 2014*). Several factors have been identified that suppress proliferation in late-pre-B cells and allow Igl recombination. These include B cell translocation gene 2 (BTG2) and protein arginine methyl transferase 1 (PRMT1) (*Dolezal et al., 2017*), the signal transducers RAS (*Mandal et al., 2009*) and dual specificity tyrosine-regulated kinase 1A (DYRK1A) (*Thompson et al., 2015*) and the transcription factors interferon regulatory factor-4 and −8 (IRF4, IRF8) (*Lu et al., 2003*), IKAROS and AIOLOS (*Ma et al., 2010*), BCL-6 (*Nahar et al., 2011*) and FOXO3a (*Herzog et al., 2008*). In pro-B cells, IL-7 promotes cell cycle progression (*Clark et al., 2014*) and the RNA-binding proteins (RBPs) ZFP36L1 and ZFP36L2 suppress proliferation, allowing Igh chain

recombination and B cell development (*Galloway et al., 2016*). Thus, by comparison to pre-B cells the mechanisms and genes that control proliferation in pro-B cells remain poorly understood.

Transcription factors act to determine which genes are transcribed and the tempo of transcription. A significant body of work has identified a network of transcription factors that control the development and identity of B cells (*Busslinger, 2004*). Amongst these, FOXO1 is essential for progression after the pro-B cell stage and to induce expression of *Rag* genes (*Dengler et al., 2008*; *Amin and Schlissel, 2008*). After transcription, numerous RBPs (*Gerstberger et al., 2014*) control messenger RNA (mRNA) expression and coordinate functionally related genes into mRNA regulons (*Keene, 2007*). These post-transcriptionally controlled networks are more challenging to identify because they may combine the effects of different RBPs or microRNAs on the splicing, polyadenylation, export, stability, localization and translation of mRNA. Dynamic gene expression during development and stress responses, which takes place on a timescale of minutes to hours, requires the coordination of transcriptional and post-transcriptional mechanisms by signalling pathways. The identity of the RBPs that regulate proliferation and differentiation of B cells remains largely unknown.

Polypyrimidine tract-binding proteins (PTBP) are RBPs with pleiotropic functions that control alternative splicing (AS), polyadenylation site-selection, mRNA stability and internal ribosome entry site (IRES)-mediated translation (*Hu et al., 2018*; *Knoch et al., 2004*; *Knoch et al., 2014*). They are encoded by a set of highly conserved paralogous genes of which PTBP1 is expressed in many cell types while PTBP2 and PTBP3 (formerly ROD1) are expressed principally in neurons and hematopoietic cells, respectively. Both PTBP1 and PTBP3 are expressed in mature B cells (*Monzón-Casanova et al., 2018*). PTBP1 can either increase or decrease mRNA stability by binding to the 3'UTR of transcripts, by modulating the AS of exons that will generate transcript isoforms that undergo degradation by nonsense-mediated mRNA decay (NMD) or by affecting polyadenylation site selection and thus the content of the 3'UTR (*Hu et al., 2018*). PTBP1 binds to *Ptbp2* mRNA and, by inducing *Ptbp2* exon 10 skipping, promotes NMD to suppress expression of PTBP2 (*Boutz et al., 2007*).

PTBP1 and PTBP2 have specific and redundant roles and expression of PTBP2 in PTBP1-deficient cells compensates for many functions of PTBP1 (*Spellman et al., 2007*; *Vuong et al., 2016*; *Ling et al., 2016*; *Monzón-Casanova et al., 2018*). To study the unique roles of PTBP1 in B cells, we and others have deleted *Ptbp1* in pro-B cells and found normal B cell development accompanied by upregulated PTBP2 expression but important defects in germinal centre (GC) responses (*Monzón-Casanova et al., 2018*; *Sasanuma et al., 2019*). In GC B cells PTBP1 promotes the selection of B cell clones with high affinity antibodies, in part by promoting the c-MYC gene expression program induced upon positive selection following T cell help, and this function is not compensated by the upregulated PTBP2 (*Monzón-Casanova et al., 2018*). Here we addressed the potential for redundancy between PTBP in B cell development. We show that PTBP3 is dispensable for B cell development while PTBP1 and, in its absence PTBP2, are essential to promote B cell lymphopoiesis beyond the pro-B cell stage. In pro-B cells, PTBP1 suppresses entry into S-phase and promotes transition into mitosis after G2-phase. At the molecular level, PTBP1 controls mRNA abundance and AS of genes important for S-phase entry and mitosis. Therefore, PTBP1 is an essential component of a previously unrecognised posttranscriptional mechanism controlling proliferation in pro-B cells.

## Results

### PTBP2 can compensate for PTBP1 in B cells

We used a panel of previously characterised PTBP paralog-specific antibodies (*Monzón-Casanova et al., 2018*) to measure the expression of the three main PTBP members at defined stages of B cell development in mouse bone marrow by flow cytometry (*Figure 1*). PTBP1 levels were similar across the different developing B cell populations as the differences in fluorescence intensity for the anti-PTBP1 antibody, where early-pre B cells had the highest fluorescence intensity levels compared to the other stages in B cell development, were also found with the isotype control staining (*Figure 1B*). PTBP2 protein was not detected in any of the B cell developmental stages analysed (*Figure 1*). PTBP3 was readily detected and also expressed at similar amounts throughout the

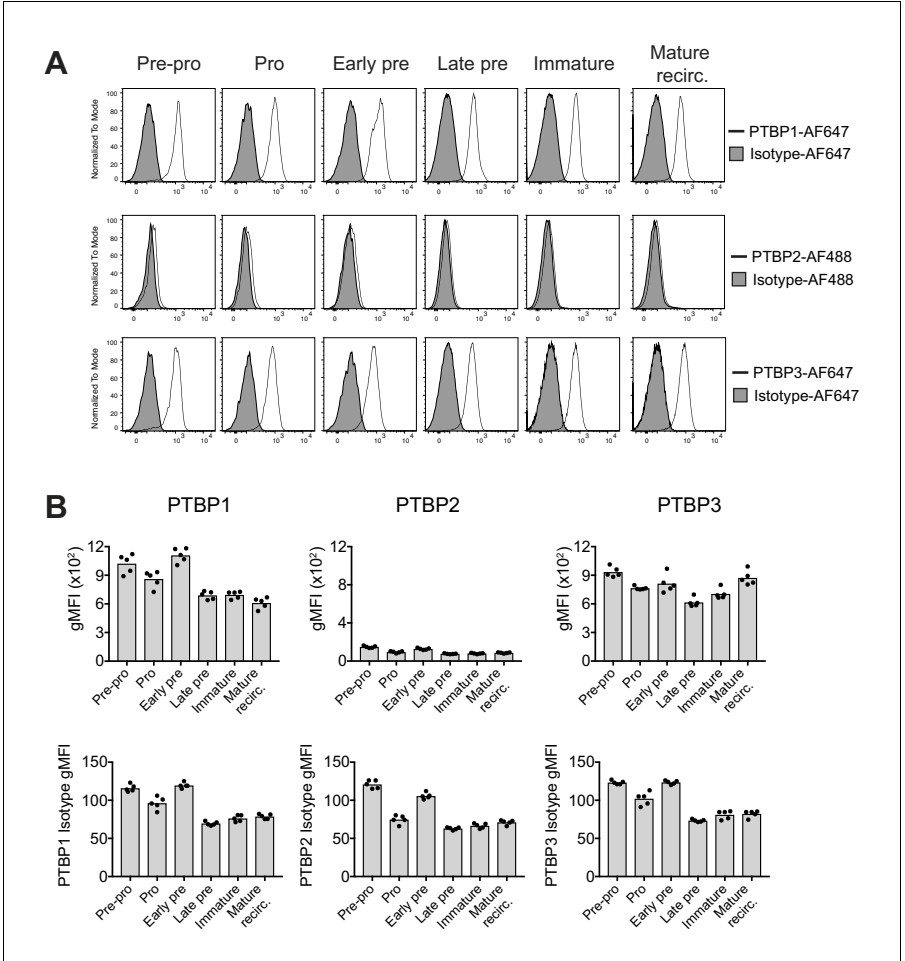

**Figure 1.** PTBP1 and PTBP3 are expressed throughout B cell development. (**A**) Expression of PTBP1, PTBP2 and PTBP3 analysed by flow cytometry. Identification of different B cell developmental stages was carried out as shown in *Figure 1—figure supplement 1B*. (**B**) Geometric mean fluorescence intensity (gMFI) of staining for anti-PTBP1, PTBP2, PTBP3 and isotype control antibodies as shown in A. Bars depict arithmetic means. Each data point shows data from an individual control mouse (*CD79a^{+/+};Ptbp1^{fl/fl};Ptbp2^{fl/fl}*). Data shown are from one experiment with five mice.

The online version of this article includes the following figure supplement(s) for figure 1:

**Figure supplement 1.** B cell developmental stages in the bone marrow.

stages of B cell ontogeny (*Figure 1*). Thus, PTBP1 and PTBP3, but not PTBP2, are expressed throughout B cell development.

To study the roles of PTBP3 in B cell development we deleted *Ptbp3* in pro-B cells with a novel *Ptbp3*-floxed allele and the *Cd79a^{cre}* allele (*Cd79a^{cre/+};Ptbp3^{fl/fl}* mice, denoted here as *Ptbp3* single conditional knock-out, P3sKO). We found that the numbers of mature B cells in the spleen of P3sKO mice were normal (*Figure 2—figure supplement 1A*). PTBP3-deficient B cells expressed PTBP1 and lacked PTBP2 (*Figure 2—figure supplement 1B*). Therefore, PTBP3 was dispensable for the development of B cells and we focused on investigating the roles of PTBP1 during B cell development.

Conditional deletion of a *Ptbp1*-floxed allele in pro-B cells mediated by a *Cd79a^{cre}* allele (*Cd79a^{cre/+};Ptbp1^{fl/fl}* mice, denoted here as *Ptbp1* single conditional knock-out, P1sKO) resulted in normal numbers of mature B cells (*Figure 2A*). Deletion of *Ptbp1* resulted in expression of PTBP2 (*Figure 2B* and *Monzón-Casanova et al., 2018*), as PTBP1 no longer promotes the degradation of *Ptbp2* transcripts by NMD (*Boutz et al., 2007*). Deletion of *Ptbp2* alone with the *Cd79a^{cre}* allele (P2sKO) had no impact on the number of mature B cells (*Figure 2A*), as expected since PTBP2 is not expressed during B cell development.

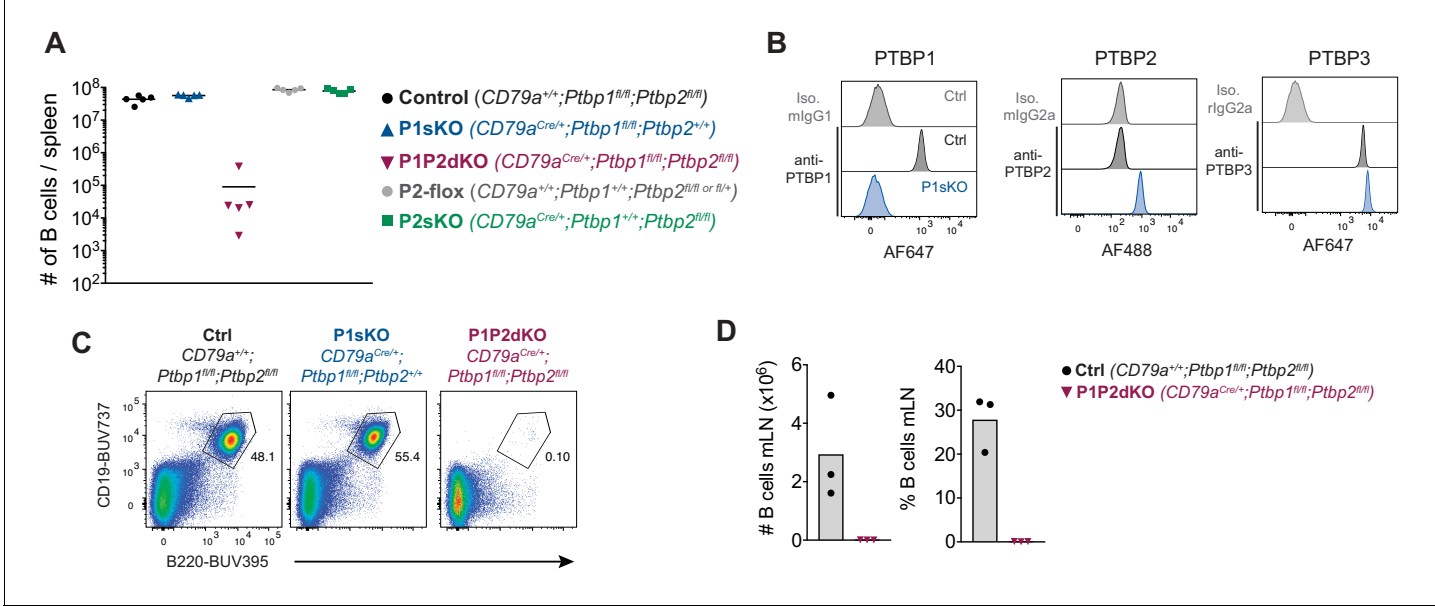

**Figure 2.** Lack of B cells in the absence of PTBP1 and PTBP2. (**A**) Numbers of B cells (B220⁺CD19⁺) in spleens of mice with the indicated genotypes. Data points are from individual mice. Arithmetic means are shown with lines. (**B**) PTBP1, PTBP2, PTBP3 and isotype control staining of splenic B cells (CD19⁺) from control mice (*CD79a^{+/+};Ptbp1^{fl/fl}*) and P1sKO (*CD79a^{Cre/+};Ptbp1^{fl/fl}*) mice analysed by flow cytometry. Data shown are from one mouse representative of three individual mice analysed. (**C**) Flow cytometry of splenocytes to identify B cells shown in A. Numbers shown are proportions of gated B cells. Events shown were pre-gated on live (eFluor780⁻) single cells. (**D**) Numbers and proportions of B cells (B220⁺CD19⁺) in mesenteric lymph nodes. Data shown are from one experiment.

The online version of this article includes the following figure supplement(s) for figure 2:

**Figure supplement 1.** Normal numbers of mature B cells in the absence of PTBP3.

To establish if PTBP2 compensated for the absence of PTBP1 we generated a double conditional knock-out (dKO) mouse model in which both *Ptbp1* and *Ptbp2* were deleted in pro-B cells using *Cd79a^{cre}* (*Cd79a^{cre/+};Ptbp1^{fl/fl};Ptbp2^{fl/fl}* mice, denoted here as P1P2dKO). P1P2dKO mice lacked mature B cells in the spleen and lymph nodes (*Figure 2A, C and D*). Thus, PTBP1 plays an essential role in the development or maintenance of mature B cells that upon PTBP1 knockout is compensated by upregulation of PTBP2.

## The essential role for PTBP1 in the absence of PTBP2 in B cell development is at the pro-B cell stage

Enumeration of cells at different stages of B cell development in the mouse bone marrow revealed that P1P2dKO mice lacked immature and mature B cells as well as small late pre-B cells (Hardy's FrD) (*Figure 3A and B*). P1P2dKO mice had slightly reduced numbers of FrB and FrC pro-B cells compared to littermate controls (*Cd79a^{+/+};Ptbp1^{fl/fl};Ptbp2^{fl/fl}*, denoted here as 'control' unless stated otherwise) and to *Cd79a^{cre/+}* 'Cre-only' mice (*Figure 3A and B*). Enumeration of pro-B cells identified as c-Kit⁺ CD19⁺ B220⁺ IgM⁻ IgD⁻ CD25⁻ cells yielded similar results (*Figure 3—figure supplement 1A*). In P1P2dKO mice FrC' early pre-B cells, characterised by high expression of CD24 and CD249 (BP-1-positive) were not detected (*Figure 3A and B*). Thus, in the absence of PTBP1 and PTBP2, B cell development is blocked at the pro-B cell stage. P1P2dKO FrC pro-B cells had higher CD24 staining than control FrC pro-B cells (*Figure 3A*). Therefore, we set the FrC gate (*Figure 3A*) in the P1P2dKO mice to include all of these cells. c-KIT (CD117) and CD2 staining showed that FrB and FrC P1P2dKO pro-B cells had high c-KIT levels and lacked CD2 (*Figure 3—figure supplement 1B*), corroborating a block at an early developmental stage. In P1sKO compared to control and *Cd79a^{cre}* 'Cre-only' mice we noticed a ~ 3 fold increase in the numbers of early-pre-B cells (*Figure 3B*). This increase is the result of higher CD43 staining in P1sKO developing B cells compared to control developing B cells (*Figure 3A*), which leads to the inclusion of more cells in the

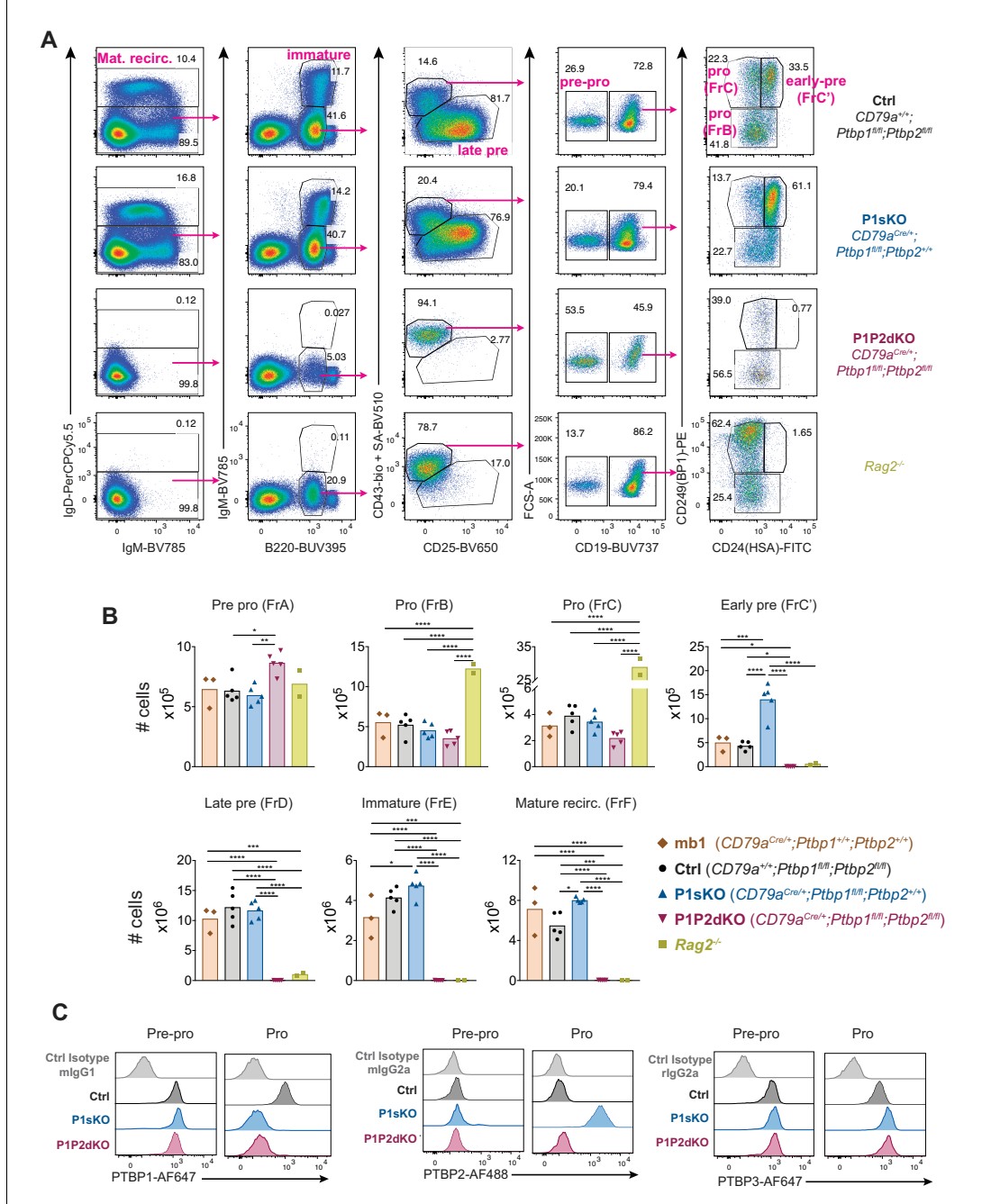

**Figure 3.** Absence of PTBP1 and PTBP2 blocks B cell development at the pro-B cell stage. (**A**) Gating strategy based on cell-surface markers for developing B cells from bone marrow cells pre-gated on dump (Gr-1, CD11b, NK1.1, Siglec-F and F4/80)-negative live (eFluor780⁻) cells. (**B**) Number of developing B cells in the bone marrow (two femurs and two tibias per mouse) of mice with the indicated genotypes. Data shown are from one representative out of three independent experiments carried out with the same mouse genotypes shown except for *CD79a^Cre/+;Ptbp1^+/+;Ptbp2^+/+* mice which were only included in the experiment shown. Bars depict arithmetic means, each point represents data from an individual mouse and P-values were calculated by one-way ANOVA with Tukey's multiple comparisons test. Summary adjusted p value *<0.05, **<0.01, ***<0.001, ****<0.0001. (**C**) PTBP1, PTBP2, PTBP3 and isotype control staining in Pre-pro and Pro-B cells identified as shown in *Figure 3—figure supplement 1C*. Data shown are of a representative mouse out of five for each indicated genotype. Data shown are from one experiment.

The online version of this article includes the following figure supplement(s) for figure 3:

**Figure supplement 1.** Absence of PTBP1 and PTBP2 blocks B cell development at the pro-B cell stage.

**Figure supplement 2.** Deletion of *Ptbp1* and *Ptbp3* results in normal B cell development in the bone marrow.

CD43 high and CD25 low pro- and early pre-B cell gate (*Figure 3B*). We compared P1P2dKO mice to *Rag2^-/-* deficient mice (*Shinkai et al., 1992*) in which B cells do not develop past the pro-B cell stage. The numbers of pro-B cells were reduced by ~4.6 fold (FrB) and 19-fold (FrC) in P1P2dKO mice compared to *Rag2^-/-* mice (*Figure 3B*). Thus, pro-B cells unable to recombine their Igh chain locus accumulated in the bone marrow whereas P1P2dKO pro-B cells did not.

To identify early pre-B cells with successfully rearranged Igμ heavy chain we used a staining strategy that included detection of intracellular Igμ (*Figure 3—figure supplement 1C*). A few early-pre-B cells (B220^+, CD19^+, IgM^-, CD93^+, CD43^high, Igμ^+) were present in P1P2dKO mice, indicating successful recombination of the Igh chain locus in P1P2dKO pro-B cells (*Figure 3—figure supplement 1C*). However, the numbers of Igμ^+ early-pre-B cells were decreased by ~13 fold in comparison to control mice (*Figure 3—figure supplement 1D*). This confirmed the block in B cell development in P1P2dKO mice at the pro-B cell stage.

P1P2dKO pro-B cells did not express PTBP1 or PTBP2 protein, demonstrating efficient gene deletion (*Figure 3C*). P1P2dKO pro-B cells expressed increased amounts of PTBP3 compared to control pro-B cells (*Figure 3C*), yet in the absence of PTBP1 and PTBP2, PTBP3 expression was insufficient to promote B cell development. Deletion of *Ptbp1* and *Ptbp3* at the pro-B cell stage with the *Ptbp1^fl*, *Ptbp3^fl* and *Cd79a^cre* alleles (P1P3dKO mice) resulted in normal numbers of developing B cells in the bone marrow including normal numbers of immature B cells (*Figure 3—figure supplement 2A*). Pro-B cells from P1P3dKO mice expressed PTBP2 and lacked both PTBP1 and PTBP3 (*Figure 3—figure supplement 2B*). Therefore, PTBP2 compensated for the absence of PTBP1 and PTBP3 during B cell development in the bone marrow. Our findings revealed high redundancy between the different PTBP paralogs which could reflect a requirement of certain amounts of total PTBP protein for the development of B cells. Importantly, we showed an essential role for PTBP1 in B cell development at the pro-B cell stage that is compensated for by PTBP2, but not by PTBP3.

## PTBP1 regulates mRNA abundance and AS in pro-B cells

To establish the genes regulated by PTBP1 necessary for successful B cell development in pro-B cells we carried out mRNAseq on cKIT^+ FrB pro-B cells from P1P2dKO, P1sKO and control mice (*Figure 4—figure supplement 1*). We carried out five biological replicates per condition and sequenced the mRNAseq libraries on a 125 bp paired-end mode obtaining ~105 million reads per sample to increase the probability of capturing reads spanning two different exons which inform on which splice sites are used. We focused our analysis on FrB pro-B cells as they are the first stage where PTBP1 and PTBP2 were depleted in B cell development with the *Cd79a^cre* allele. Thereby, we increased the likelihood of identifying direct targets of PTBP1 amongst the genes with differences in gene expression. To identify transcripts directly bound by PTBP1 we made use of a PTBP1 individual-nucleotide resolution Cross-Linking and ImmunoPrecipitation (iCLIP) dataset (*Monzón-Casanova et al., 2018*) which reveals PTBP1-binding sites in the whole transcriptome and thereby allows distinction between direct and indirect targets. The PTBP1 iCLIP data were from mitogen-activated mouse primary B cells because it was not feasible to purify sufficient numbers of pro-B cells for iCLIP. There is a positive correlation between the transcriptomes of mitogen-activated primary B cells and pro-B cells (*Figure 4—figure supplement 2A*), suggesting that the PTBP1 iCLIP data set (*Supplementary files 1* and *2*) is suitable to infer PTBP1-bound RNAs in pro-B cells. We only found 681 genes with no mRNA expression (0 Transcripts Per Million (TPM) in activated primary B cells that were expressed in pro-B cells (>=1 TPM). Therefore, the probability of missing specific targets of PTBP1 in pro-B cells using the iCLIP data from mature activated B cells was small.

We analysed changes in mRNA abundance by comparing pro-B cell transcriptomes from the different genotypes in pairwise comparisons (*Figure 4A*). More genes showed differential mRNA abundance when comparing P1P2dKO to control pro-B cells and P1P2dKO to P1sKO pro-B cells than when comparing P1sKO to control pro-B cells (*Figure 4A* and *Figure 4—source data 1*). Almost one-quarter of the genes with increased or decreased mRNA abundance in P1P2dKO pro-B cells encoded transcripts that were directly bound by PTBP1 at the 3'UTR (*Figure 4—figure supplement 2B* and *Figure 4—source data 1*). The remaining changes observed in mRNA abundance could be attributed to indirect effects, or to roles of PTBP1 in controlling transcript abundance that are independent of 3'UTR binding such as AS leading to NMD (AS-NMD). A striking example of the latter effect is that *Ptbp2* mRNA abundance was increased ~18 fold in P1sKO compared to control pro-B

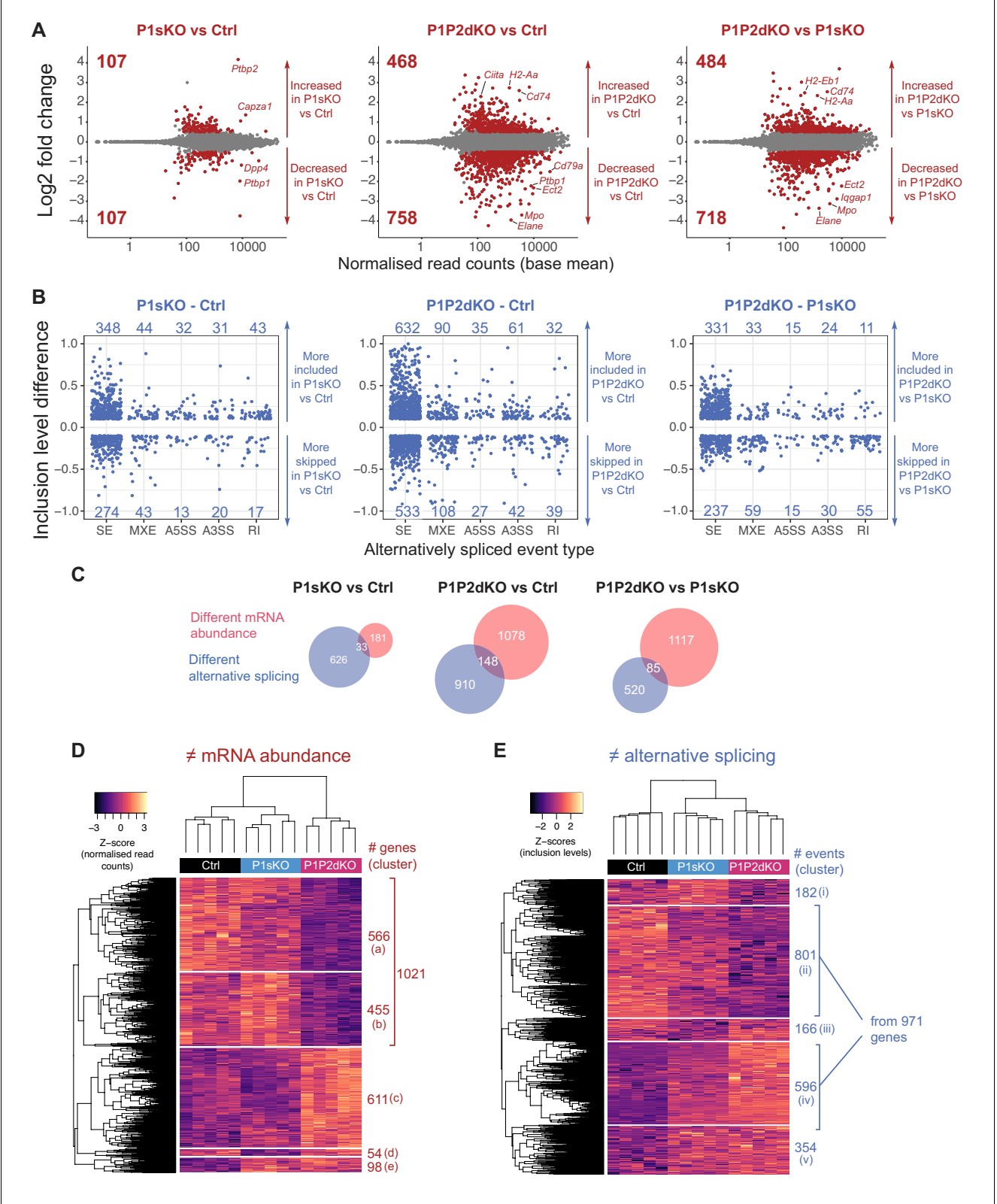

**Figure 4.** PTBP1 and PTBP2 absence causes changes in mRNA abundance and AS. (A) Differences in mRNA abundance in pairwise comparisons from pro-B cell transcriptomes. Shown are Log2Fold changes calculated with DESeq2 (*Figure 4—source data 1*). Red dots are values from genes with significant differences in mRNA abundance (padj value <0.05) with a |Log2Fold change| > 0.5. Grey dots are values from genes with no significant differences (padj >0.05) or with a |Log2Fold change| < 0.5. Numbers in plots are the number of genes with increased or decreased mRNA abundance.
*Figure 4 continued on next page*

*Figure 4 continued*

(B) Differences in AS in pairwise comparisons from pro-B cells. Shown are |inclusion level differences| > 0.1 with an FDR < 0.05 for different types of alternatively spliced events (*Figure 4—figure supplement 2C*): skipped exons (SE), mutually exclusive exons (MXE), alternative 5' and 3' splice sites (A5SS and A3SS, respectively), and retained introns (RI), (*Figure 4—source data 2*) analysed with rMATS. Each dot shows the inclusion level difference for one event. (C) Overlaps of genes that have changes in abundance and AS. (D) Heatmap shows z-scores of normalised read counts from DESeq2 from each biological replicate for genes that were found with differential mRNA abundance in any of the three pair-wise comparisons shown in A. (E) Heatmap shows z-scores of inclusion levels from each biological replicate for those AS events that are alternatively spliced in any of the three pairwise comparisons shown in B. (D, E) Unsupervised hierarchical clustering was done on Euclidean distances from z-scores and is shown with dendrograms. The online version of this article includes the following source data and figure supplement(s) for figure 4:

**Source data 1.** Changes in mRNA abundance.
**Source data 2.** Changes in AS.
**Figure supplement 1.** Cell sorting strategy of pro-B cells.
**Figure supplement 2.** Transcriptome analysis of pro-B cells.

cells (*Figure 4A*) since *Ptbp2* transcripts in P1sKO pro-B cells included exon 10 and were no longer degraded by NMD.

We used rMATS to compute the differences in exon inclusion levels in pairwise comparisons of the genotypes (*Shen et al., 2014*; *Figure 4B* and *Figure 4—figure supplement 2C*). The inclusion level of a particular alternatively spliced event is displayed as a proportion, scaled to a maximum of 1, of transcripts containing the alternatively spliced mRNA segment. Similar to our observations on mRNA abundance, the absence of both PTBP1 and PTBP2 in pro-B cells resulted in more changes in AS than the absence of PTBP1 (*Figure 4B* and *Figure 4—source data 2*). From 30% to 50% of the events with AS changes when comparing P1P2dKO to control pro-B cells were bound by PTBP1 (*Figure 4—figure supplement 2D* and *Figure 4—source data 2*) implicating PTBP1 in controlling these events directly. There is a small overlap of genes with changes in mRNA abundance and also AS in the different pairwise comparisons (*Figure 4C*), suggesting that the inclusion of certain exons generates NMD targets. Such AS events promoting NMD will be underestimated in our data, since NMD-targeted isoforms will be degraded and difficult to detect by mRNAseq.

As B cell development is largely unaffected in P1sKO mice, we sought to identify differences in mRNA abundance and AS unique to P1P2dKO pro-B cells compared to both P1sKO and control pro-B cells to identify the changes causing the block in B cell development in the absence of PTBP1 and PTBP2. To this end we carried out unsupervised clustering of the genes with differential mRNA abundance between the different genotypes and genes that were differentially spliced. We identified 1021 and 611 genes with decreased and increased mRNA abundance, respectively, in P1P2dKO compared to control and P1sKO pro-B cells (*Figure 4D*). Amongst AS changes, we found 971 genes with increased or decreased inclusion levels of at least one event in P1P2dKO pro-B cells compared to control and P1sKO (*Figure 4E*). P1P2dKO pro-B cells clustered separately from P1sKO and control pro-B cells when considering abundance changes (*Figure 4D*). When analysing AS, P1P2dKO and P1sKO pro-B cells clustered together but separately from control pro-B cells (*Figure 4E*). Therefore, the compensatory functions of PTBP2 when PTBP1 is absent were more evident when considering mRNA abundance than AS. We observed only 54 genes with different mRNA abundance (cluster d) or changes in AS (clusters i and iii, 384 events) that were predominantly regulated in P1sKO pro-B cells compared to P1P2dKO and control pro-B cells (*Figure 4D and E*). In contrast, large numbers of genes had different mRNA abundance (clusters a, b and c, 1632 genes) and AS (clusters ii and iv, 1397 events) in P1P2dKO pro-B cells compared to P1sKO and control pro-B cells (*Figure 4D and E*). Therefore, in pro-B cells, PTBP2 has only a few specific targets and mostly compensates for the absence of PTBP1, while PTBP1 ensures the appropriate expression at the level of AS and mRNA abundance of more than 2000 genes.

## PTBP1 regulates pathways associated with growth and proliferation

To understand the roles of PTBP1 in B cell development we first inspected mRNA expression of genes important for B cell lymphopoiesis. We found similar mRNA abundance in P1P2dKO compared to control and P1sKO pro-B cells in most of these genes including *Cnot3* (*Inoue et al., 2015*; *Yang et al., 2016*) and *Pax5* (*Urbánek, 1994*; *Figure 4—figure supplement 2E*). Similarly, genes regulating cell survival and apoptosis such as *Bcl2*, *Bcl2l2* (*Bcl-XL*), *Bax* and *Bcl2l11* (*Bim*) had normal

mRNA abundance and AS patterns in P1P2dKO pro-B cells (*Figure 4—source datas 1* and *2*). E2A (encoded by *Tcf3*) and IKAROS are two transcription factors with AS (*Schjerven et al., 2013*; *Sun and Baltimore, 1991*) important for B cell ontogeny. In human, PTBP1 regulates AS of *Tcf3* (*Yamazaki et al., 2019*). We found similar AS patterns of *Ikaros* and *Tcf3* in P1P2dKO compared to control pro-B cells (*Figure 4—source data 2*). *Ebf1* mRNA abundance was reduced ~1.5 fold in P1P2dKO compared to control and P1sKO pro-B cells (*Figure 5A*), but this should not impact B cell development because *Ebf1* haploinsufficiency results in normal B cell ontogeny (*Vilagos et al., 2012*; *Győry et al., 2012*). *Foxo1* mRNA abundance was reduced ~1.7 fold in P1P2dKO compared to P1sKO and control pro-B cells (*Figure 5A*). PTBP1 bound directly to the 3'UTR of *Foxo1* (*Figure 5B*), indicating a direct role of PTBP1 stabilising *Foxo1* mRNA. *Foxo1* deficient pro-B cells do not develop further and have reduced IL-7 receptor expression (*Dengler et al., 2008*). Indeed, *Il7r* mRNA abundance was reduced ~1.4 fold in P1P2dKO compared to P1sKO and control pro-B cells (*Figure 5A*). However, IL-7 receptor staining was similar between P1P2dKO and control pro-B cells (*Figure 5C*), ruling out a role for a reduction of IL-7 receptor in the developmental defect of P1P2dKO pro-B cells. FOXO1 promotes RAG expression (*Amin and Schlissel, 2008*) but *Rag1* and *Rag2* mRNA abundance and AS were unaffected in P1P2dKO pro-B cells (*Figure 4—figure supplement 2E* and *Figure 4—source data 2*). These data indicate that the known essential elements of the B cell development programme, including three targets of FOXO1, are independent of the PTBP.

To assess additional cellular pathways affected in the absence of PTBP1 and PTBP2, we carried out gene ontology (GO) enrichment analysis with the genes that showed altered mRNA expression in P1P2dKO compared to P1sKO and control pro-B cells. We analysed genes with increased or decreased mRNA abundance and changes in AS separately (*Figure 5D* and *Figure 5—source data 1*). We found numerous enriched GO terms important for the biology of B cells and their progenitors (*Figure 5D* and *Figure 5—source data 1*) such as 'antigen processing and presentation of peptide antigen via MHC class II' amongst genes with increased mRNA abundance and 'response to lipopolysaccharide' amongst genes with decreased mRNA abundance. 'Regulation of neuron differentiation' was enriched amongst genes with reduced abundance and with changes in AS, as we expected from the known roles of PTBPs in neuronal development (*Hu et al., 2018*). Amongst genes with increased mRNA abundance we also found an enrichment for 'ribosome biogenesis', 'ribonucleoside biosynthetic process' and 'rRNA processing', indicating a higher biosynthetic capacity of P1P2dKO pro-B cells compared to control and P1sKO pro-B cells.

Amongst genes with reduced mRNA abundance there was an enrichment for 'cyclin-dependent protein serine/threonine kinase inhibitor activity'. Cyclin dependent kinase (CDK) inhibitors are major regulators of cell cycle progression (*Otto and Sicinski, 2017*; *Yoon et al., 2012*) and could be relevant in controlling the proliferation of P1P2dKO pro-B cells. We found that P1P2dKO FrB pro-B cells had a reduced mRNA abundance of *Cdkn1b*, *Cdkn1c*, *Cdkn2c* and *Cdkn2d* (encoding p27, p57, p18 and p19, respectively) compared to control and P1sKO FrB pro-B cells (*Figure 5E*). None of these CDK inhibitors has an obvious change in AS or a binding site for PTBP1 in the 3'UTR (*Figure 4—source datas 1* and *2*) although they were expressed in mature B cells (data not shown). FOXO1 promotes p27 expression (*Nakamura et al., 2000*). Therefore, reduced FOXO1 expression (*Figure 5A*) could result in the observed reduced *Cdkn1b* mRNA abundance (*Figure 5E*) in P1P2dKO pro-B cells. Thus, the reduction in mRNA abundance of CDK inhibitors is likely to be an indirect effect of the lack of PTBP1 and PTBP2 at least partly mediated through direct FOXO1 regulation.

## PTBP1 represses the entry of pro-B cells into S-phase

The enrichment for CDK inhibitors amongst genes with reduced mRNA abundance and the increased mRNA abundance of biosynthetic pathway components in P1P2dKO compared to P1sKO and controls prompted us to assess the proliferative status of pro-B cells in the P1P2dKO mice. We first assessed the proportions of pro-B cells in S-phase by detecting cells that had incorporated EdU for one hour in vivo (*Figure 6—figure supplement 1*). We found an increase in the proportions of FrB and FrC pro-B cells that are in S-phase (2-fold in FrB and 4-fold in FrC) in P1P2dKO compared to control and P1sKO mice (*Figure 6—figure supplement 1A and B*). Moreover, G0/G1 P1P2dKO pro-B cells had increased FSC-A measurements compared to P1sKO and control G0/G1 pro-B cells

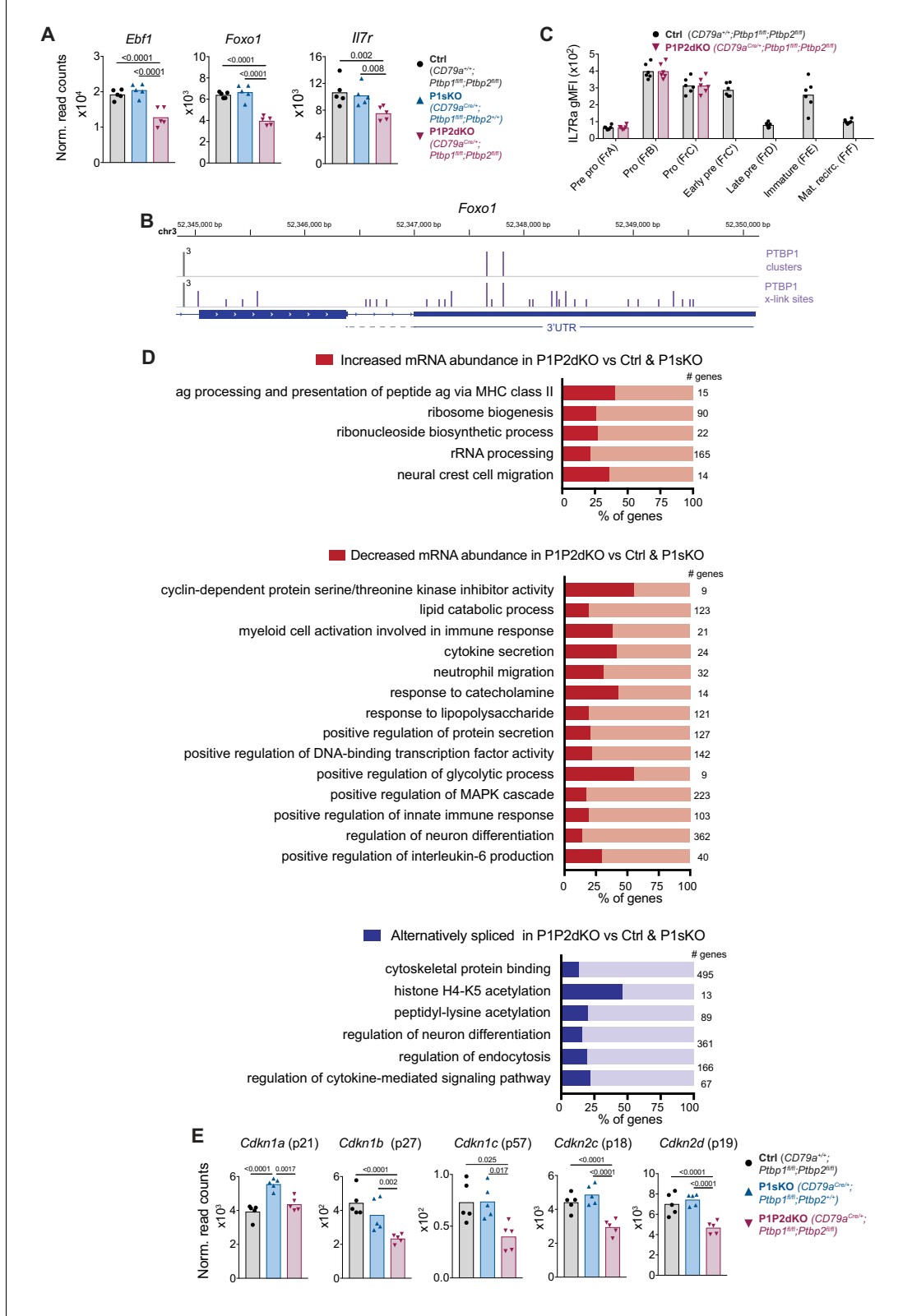

**Figure 5.** PTBP1 regulates pathways associated with growth and proliferation. (**A**) *Ebf1*, *Foxo1* and *IL7r* mRNA abundance in FrB pro-B cells from control, P1sKO and P1P2dKO mice. (**B**) PTBP1 binding (iCLIP data) to the *Foxo1* 3'UTR. (**C**) IL-7R (CD127) geometric mean fluorescent intensity in stages of B cell development identified as shown in *Figure 3A*. Individual data points are from individual mice. Data shown are from one representative out of two experiments. Unpaired T-test was carried out comparing control and P1P2dKO developing B cells. p values were > 0.05 and are not shown. (**D**)

*Figure 5 continued*

Selected gene ontology (GO) terms (process and function) significantly (p-value<0.05) enriched amongst genes that are differentially expressed at the abundance or the AS level when comparing the transcriptome of P1P2dKO pro-B cells to P1sKO and control pro-B cells as shown in *Figure 4D and E*. Numbers show how many genes a GO term relates to. *Figure 5—source data 1* contains all significantly enriched GO process and function terms. (E) mRNA abundance of CDK inhibitors in pro-B cells from control, P1sKO and P1P2dKO mice. (A and E) Points show DESeq2 normalised read counts from individual pro-B cell mRNAseq libraries. Bars depict arithmetic means. DESeq2 calculated p-adjusted values are shown when < 0.05 for the indicated pairwise comparisons.

The online version of this article includes the following source data for figure 5:

**Source data 1.** Gene ontology enrichment analysis.

(*Figure 6—figure supplement 1C*), which are indicative of an increased cell size and are consistent with a high biosynthetic capacity and a predisposition to enter S-phase.

To understand if the increased proportions of P1P2dKO pro-B cells in S-phase were due to an enhanced entry into S-phase or due to an accumulation in S-phase, we sequentially labelled cells in vivo with EdU and BrdU (*Figure 6A*). With this approach, we identified cells in early-, late- or post-S-phase in the whole pro-B cell population. Cells that incorporated only EdU (post-S-phase) were in S-phase at the beginning, but not at the end of the labelling period; cells that incorporated both EdU and BrdU were in late S-phase; and cells labelled only with BrdU at the end of the labelling period were in early S-phase. In control mice, we detected few pre-pro-B cells in S-phase (3.8% in early- and late-S), compared to 13% of pro-B cells and 53% of early-pre-B cells (*Figure 6—figure supplement 2B and C*). In P1P2dKO pro-B cells the proportions of early-S-phase cells were increased 4.7-fold compared to control and 3.2-fold compared to P1sKO pro-B cells (*Figure 6B and C*). The proportions of P1P2dKO pro-B cells in late S-phase and that had exited S-phase (post S-phase) were also increased compared to P1sKO pro-B cells and control pro-B cells (*Figure 6C*). The magnitude of the increase in the proportions of cells in post S-phase was smaller than the increase in either early- or late-S-phase (*Figure 6C*). This indicated that P1P2dKO pro-B cells enter S-phase more readily than control pro-B cells but they fail to progress through S-phase at the same pace as PTBP1-sufficient pro-B cells.

## PTBP1 promotes the entry of pro-B cells into mitosis

To look for possible additional effects of PTBP1 upon the G2/M-phases of the cell cycle, we analysed DNA synthesis in combination with DNA content (*Figure 6D*). Compared to control and P1sKO, P1P2dKO pro-B cells in G2/M-phase were increased both in proportion (~40 fold) and number (~8 fold) (*Figure 6E*). These findings were confirmed when identifying FrB and FrC pro-B cells using cell surface markers (CD19$^+$, CD24$^+$, CD249$^-$, CD2$^-$, CD25$^-$, IgM$^-$, IgD$^-$) (*Figure 6—figure supplement 1A and B*). To distinguish G2 and M phases of the cell cycle we stained FrB pro-B cells both for DNA and phosphorylated histone 3 (pH3) which marks cells in mitosis (*Figure 6F*). The proportion of pH3-positive cells amongst P1P2dKO FrB pro-B cells with 4N-DNA content was reduced ~3 fold compared to the proportions found amongst P1sKO and control FrB pro-B cells (*Figure 6G*). Therefore, the majority of P1P2dKO pro-B cells with 4N-DNA content were in G2-phase and had not entered mitosis. PTBP1, and in its absence PTBP2, were thus required for pro-B cells to progress from G2 to M-phase of the cell cycle.

## PTBP1 controls CDK activity in pro-B cells

The decrease in mRNA abundance of CDK inhibitors (*Figure 5E*) and the abnormal proliferation observed in pro-B cells due to the absence of PTBP1 and PTBP2 (*Figure 6*) prompted us to measure expression of the CDK inhibitor p27 in the different phases of the cell cycle. We found p27 staining reduced ~1.7 fold in G0/G1 P1P2dKO FrB pro-B cells compared to G0/G1 P1sKO and control FrB pro-B cells (*Figure 7A and B*), consistent with the decreased *Cdkn1b* mRNA abundance in FrB pro-B cells (*Figure 5E*). In contrast, in G2 cells p27 staining was increased ~7 fold in P1P2dKO FrB pro-B cells compared to P1sKO and control FrB pro-B cells (*Figure 7A and B*). These expression patterns of p27 in P1P2dKO FrB pro-B cells were conserved in P1P2dKO FrC pro-B cells (*Figure 7—figure supplement 1*). p27 expression is regulated at the post-translational level as well as the transcriptional level (*Vervoorts and Lüscher, 2008*). Moreover, PTBP1 promotes p27 IRES-mediated translation directly in human cells (*Cho et al., 2005*). We did not find mouse PTBP1 bound to the *Cdkn1b*

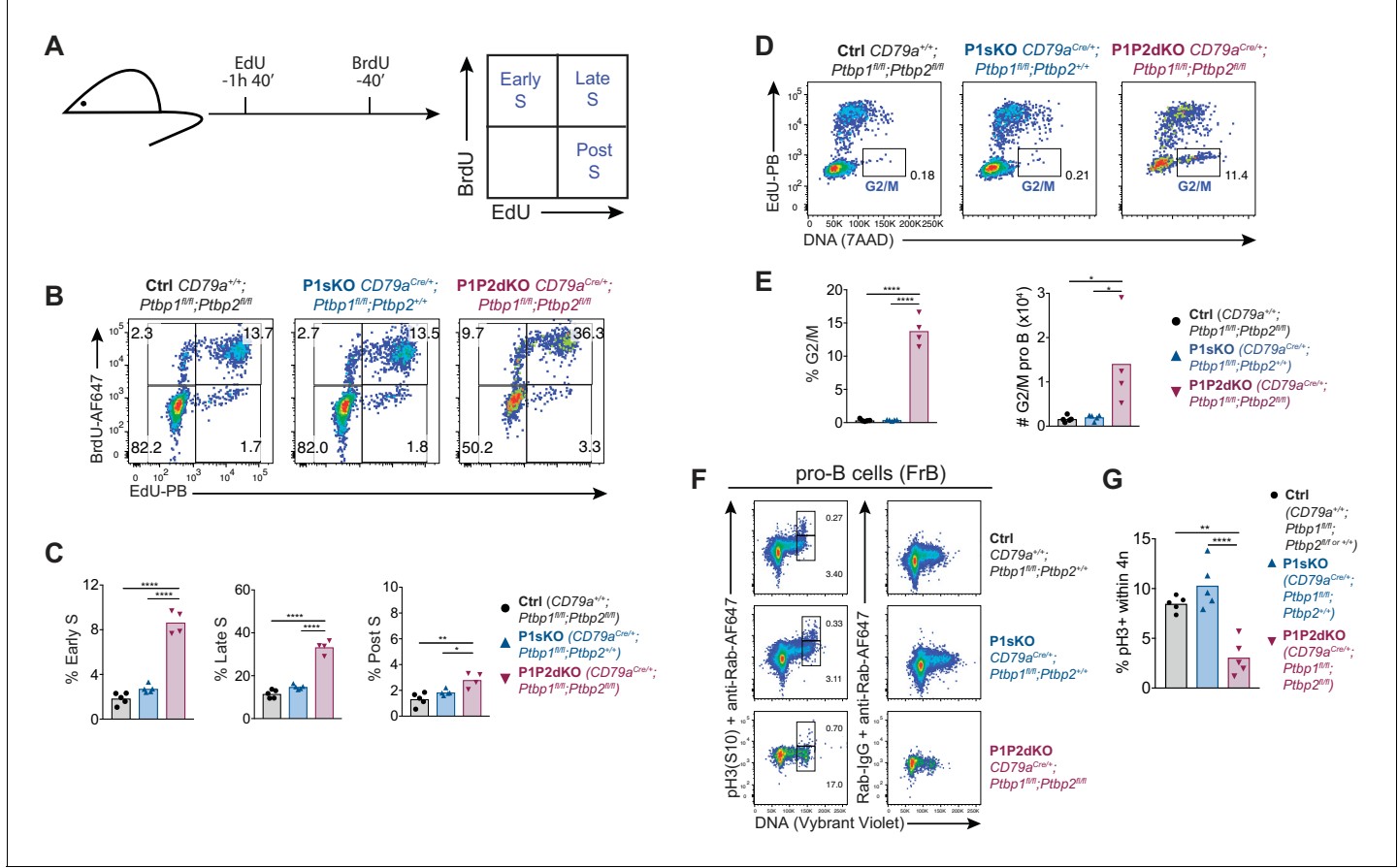

**Figure 6.** Enhanced entry into S-phase and block at G2 in P1P2dKO pro-B cells. (A) EdU and BrdU sequential labelling experimental set up to distinguish early, late and post S-phase cells. (B) Flow cytometry data of the different stages of S-phase in pro-B cells (B220+CD19+IgD-surfaceIgM-intracellular-Igµ-CD43high) identified as shown in *Figure 6—figure supplement 2A*. Numbers shown are proportions of cells. (C) Percentages of pro-B cells in different S-phase stages determined as shown in A and B. (D) Flow cytometry data of pro-B cells identified as shown in *Figure 6—figure supplement 2A* and excluding BrdU+-only (cells in early S-phase). Numbers shown are proportions of cells in the G2/M gate. (E) Proportions and numbers (in 2 femurs and two tibias per mouse) of pro-B cells in G2/M identified as shown in (D). (F) Phospho-histone 3 serine 10 (pH3 S10) staining amongst FrB pro-B cells identified as in *Figure 6—figure supplement 1A*. (G) Percentages of pH3(S10)-positive cells amongst cells with 4N DNA amounts in pro-B cells (FrB) assessed by flow cytometry as shown in F. (C, E, G) Bars depict arithmetic means, each point represents data from an individual mouse and P-values were calculated by one-way ANOVA with Tukey's multiple comparisons test. Summary adjusted p-value *<0.05, **<0.01, ***<0.001, ****<0.0001. (B–E) Data shown are from one of two independent experiments. (F, G) Data shown are from one out of three independent experiments.

The online version of this article includes the following figure supplement(s) for figure 6:

**Figure supplement 1.** Cell cycle analysis in FrB and FrC pro-B cells.

**Figure supplement 2.** Cell cycle analysis in pro-B cells.

5'UTR (*Supplementary file 2*) although *Cdkn1b* mRNA is expressed in mitogen-activated B cells. Thus, additional layers of regulation are expected to contribute to the dynamic regulation of p27 protein abundance across the different phases of the cell cycle observed in the absence of PTBP1 and PTBP2.

In addition to p27 expression, we assessed CDK activity by measuring the extent of RB phosphorylation in Ser-807/811, which is mediated in G1 by CDKs and promotes the entry into S-phase (*Otto and Sicinski, 2017*), and of SAMHD1 phosphorylation in Thr-592, which is a surrogate marker for CDK1 activity (*Cribier et al., 2013*). Proportions of both p-RB and p-SAMHD1-positive cells were increased in FrB and FrC P1P2dKO G0/G1 pro-B cells compared to control and P1sKO G0/G1 pro-B cells (*Figure 7C and D*, and *Figure 7—figure supplement 1B*) indicating an abnormally high CDK activity in G0/G1 P1P2dKO pro-B cells. In contrast, amongst pro-B cells in G2/M-phases the

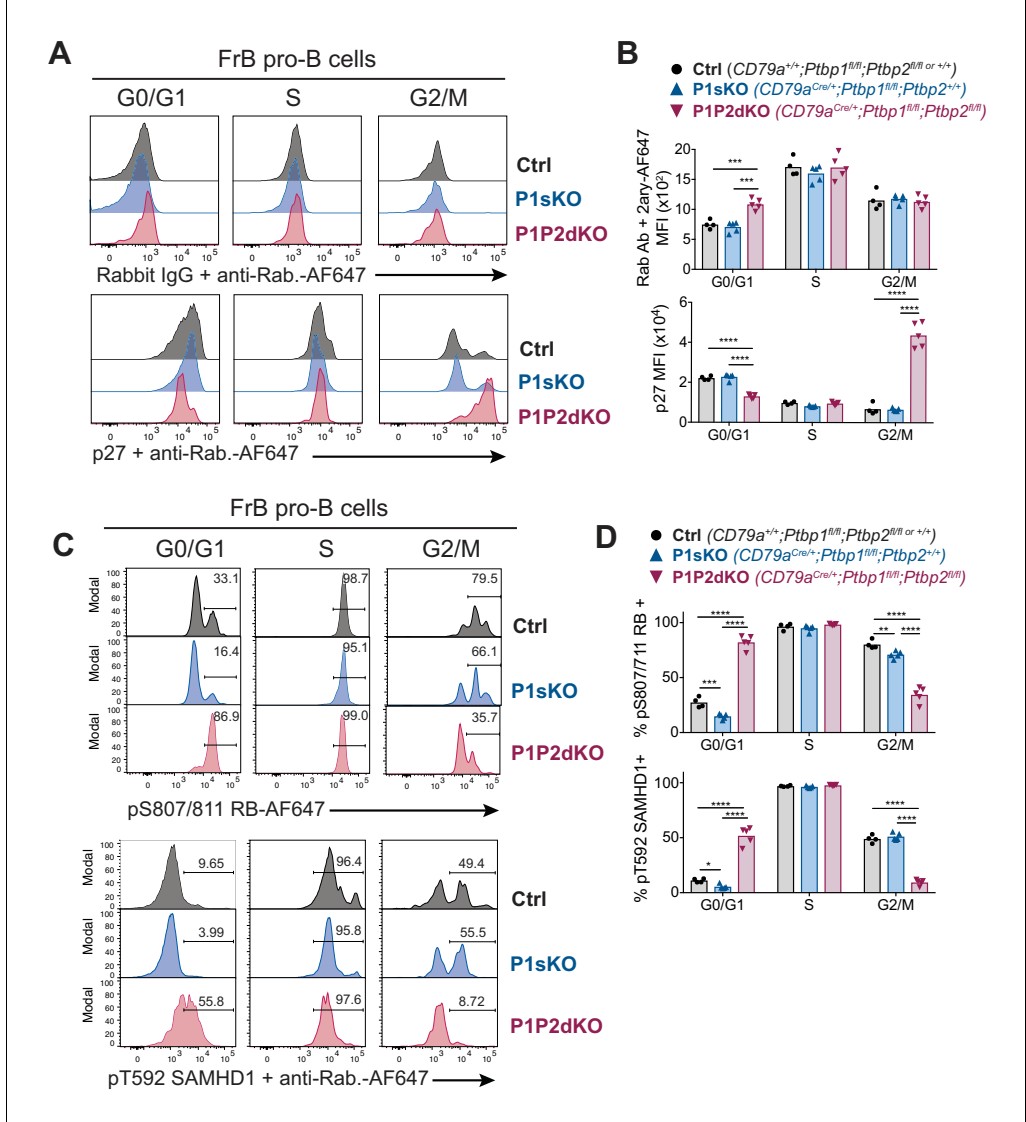

**Figure 7.** PTBP1 controls CDK activity in FrB pro-B cells. (A) Intracellular flow cytometry with anti-p27 antibody or control isotype staining detected with an anti-rabbit AF647-conjugated secondary antibody. (B) Median fluorescence intensities (MFI) from staining shown in A. (C) Intracellular flow cytometry with the indicated antibodies. Numbers show proportions of gated events. (D) Proportions of cells identified in C. (A, C) Each histogram line shows data from an individual mouse with the indicated genotype. FrB pro-B cells in G0/G1, S or G2/M phases of the cell cycle were defined by EdU incorporation and DNA staining as shown in *Figure 6—figure supplement 1A*. (B, C) Points show data from individual mice. Bars depict arithmetic means. P-values were calculated by two-way ANOVA with Tukey's multiple comparisons test. Summary adjusted p value *<0.05, **<0.01, ***<0.001, ****<0.0001. (A–D) Data are from one experiment with four to five mice per genotype. The differences observed for pT592-SAMHD1 and pS807/S811-RB between control and P1P2dKO cells were confirmed in an independent experiment where two control and two P1P2dKO mice were used.

The online version of this article includes the following figure supplement(s) for figure 7:

**Figure supplement 1.** PTBP1 controls CDK activity in FrC pro-B cells.

proportions of p-RB and p-SAMDH1-positive cells were reduced in P1P2dKO mice compared to control and P1sKO mice (*Figure 7C and D* and *Figure 7—figure supplement 1B*). This indicates a requirement for PTBP1 to achieve sufficient CDK1 activity to enter mitosis. Thus, PTBP1 is essential for controlling the activity of CDKs in different phases of the cell cycle in pro-B cells, to limit entry into S-phase and to promote entry into mitosis.

## PTBP1 controls the expression of S-phase entry regulators in pro-B cells

GO enrichment analysis of genes differentially expressed in P1P2dKO compared to control and P1sKO pro-B cells identified an enrichment for ribosome biogenesis and CDK inhibitors but it did not detect further terms directly related with proliferation. Therefore to identify PTBP1 targets directly implicated in the progression through different phases of the cell cycle we assessed changes in mRNA abundance of genes known to be highly expressed specifically in the S and G2/M-phases (*Giotti et al., 2019*) between FrB pro-B cells of the different genotypes. S- and G2/M-associated transcripts were not globally increased in P1P2dKO FrB pro-B cells compared to control and P1sKO FrB pro-B cells (*Figure 8A* and *Figure 8—source data 1*). This indicates that the majority of the changes observed in the transcriptome of P1P2dKO compared to control and P1sKO pro-B cells are not the result of comparing populations with different proliferative status, but are reflective of an altered pro-B cell transcriptome. Thus, instead of gene sets, it was possible that individual genes with rate limiting properties for cell cycle progression, which could also control the expression of CDK-inhibitors and CDK activities, were directly regulated by PTBP1. Therefore, we considered further genes affected in the absence of PTBP1 and PTBP2 but not in the absence of PTBP1 alone that are known to regulate the cell cycle.

We identified four genes associated with S-phase entry with direct PTBP1 binding and changes at the levels of mRNA abundance and AS. Two genes (*Myc* and *Ccnd2*) whose proteins promote entry into S-phase, showed increased mRNA abundance in P1P2dKO compared to P1sKO and control FrB pro-B cells (*Figure 8B*). Two further genes (*Btg2* and *Rbl1*), whose proteins inhibit S-phase entry, had reduced mRNA abundance in P1P2dKO compared to control and P1sKO FrB pro-B cells (*Figure 8B*). PTBP1 bound to the 3'UTR of *Myc* (*Figure 8—figure supplement 1*), and we confirmed an increase of c-MYC protein abundance in G0/G1 P1P2dKO FrB pro-B cells compared to control and P1sKO G0/G1 FrB pro-B cells (*Figure 8C and D*) as the fold-change increase in fluorescence intensity of the c-MYC staining (2.2-fold) was greater than the fold-change increase in fluorescence intensity of the control staining (1.5-fold). FrC P1P2dKO pro-B cells in G0/G1 also showed increased cMYC-staining compared to control FrC pro-B cells (*Figure 8—figure supplement 2*). Therefore, in pro-B cells PTBP1 suppresses c-MYC.

CYCLIN-D2 pairs with CDK4/6 and promotes entry into S-phase (*Otto and Sicinski, 2017*). *Ccnd2* abundance (encoding CYCLIN-D2) was increased ~2 fold in P1P2dKO compared to control and P1sKO FrB pro-B cells (*Figure 8B*) and PTBP1 bound to the *Ccnd2* 3'UTR (*Figure 8—figure supplement 1*). c-MYC induces transcription of *Ccnd2* (*Bouchard et al., 2001*). Therefore, PTBP1 and PTBP2 could reduce *Ccnd2* mRNA abundance indirectly by suppressing c-MYC-mediated transcription and directly by binding its mRNA.

BTG2 inhibits G1 to S transition and promotes B cell development by suppressing proliferation in late-pre-B cells (*Dolezal et al., 2017*). *Btg2* mRNA abundance was reduced ~2 fold due to *Ptbp1* and *Ptbp2* deletion in FrB pro-B cells (*Figure 8B*). No obvious change in *Btg2* AS was detected but PTBP1 binding sites were present at the *Btg2* 3'UTR (*Figure 8—figure supplement 1*). Therefore, PTBP1 could directly promote BTG2 expression in FrB pro-B cells by stabilising *Btg2* transcripts. p107 (encoded by *Rbl1*) represses E2F transcription factors and entry into S-phase (*Bertoli et al., 2013*). *Rbl1* mRNA abundance was reduced ~1.5 fold in P1P2dKO compared to control and P1sKO pro-B cells (*Figure 8B*). PTBP1 and PTBP2 suppress the inclusion of a downstream alternative 5' splice site (A5SS) in *Rbl1* exon 8. Use of this A5SS in P1P2dKO FrB pro-B cells generates an NMD-target isoform (*Figure 8E*). PTBP1 binds to adjacent regions of this alternatively spliced event (*Figure 8E*). This A5SS event is conserved in human, as PTBP1- and PTBP2-depleted HeLa cells (*Ling et al., 2016*) also have an increased usage of the downstream A5SS compared to PTBP-sufficient cells, resulting in reduced *RBL1* mRNA abundance (*Figure 8F and G*). Therefore, PTBP1 most likely promotes *Rbl1* expression by suppressing production of the NMD-targeted *Rbl1* isoform. Collectively, deregulated expression of *Myc*, *Ccnd2*, *Btg2* and *Rbl1* in the absence of PTBP1 and PTBP2 would act to drive entry of pro-B cells into S-phase.

## PTBP1 promotes expression of a network of mitotic factors via AS

We also identified transcripts important for the transition from G2 to M-phase with altered expression in P1P2dKO FrB pro-B cells (*Figure 8A*). The abundance of *Cdc25b*, *Ect2*, *Kif2c* and *Kif22* was reduced (~3,~5,~2 and~2 fold, respectively) in P1P2dKO compared to control and P1sKO FrB pro-B

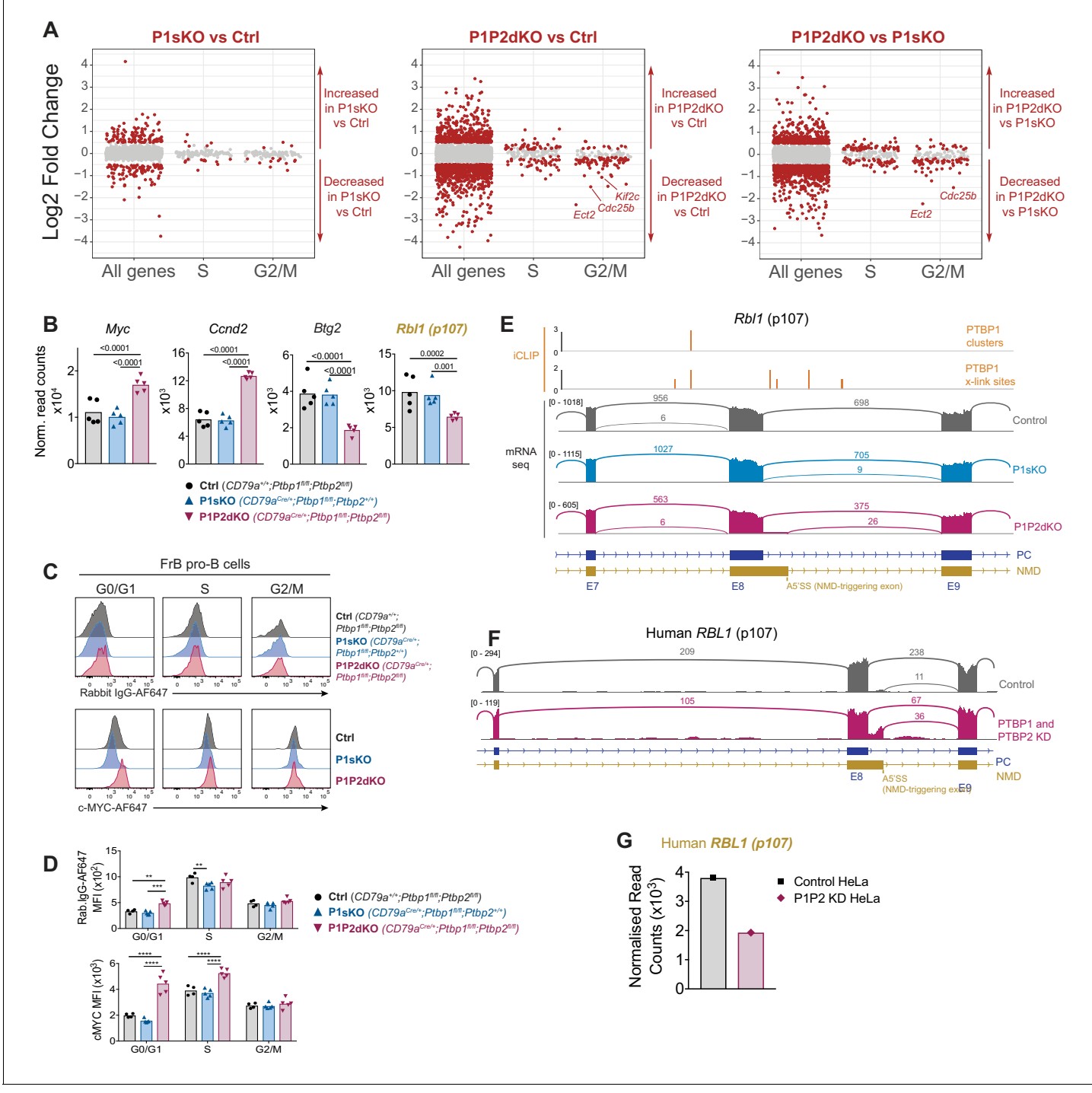

**Figure 8.** PTBP1 controls expression of genes important for S-phase entry. (**A**) Log2-fold changes in mRNA abundance in the indicated pairwise comparisons for all tested genes (all genes), genes with increased abundance in S-phase (S) and G2/M-phases (G2/M). Grey dots show genes with padj >0.05. Red dots amongst 'all genes' show genes with padj <0.05 and a |log2-fold change| > 0.5. Red dots amongst S and G2M groups show genes with a padj <0.05 regardless of their log2-fold change. (**B**) mRNA abundance in pro-B cells from control, P1sKO and P1P2dKO mice. *Rbl1* (name in yellow) is predicted to have reduced mRNA abundance due to changes in AS triggering NMD upon *Ptbp1* and *Ptbp2* deletion. Individual data points show DESeq2 normalised read counts from individual pro-B cell mRNAseq libraries. Bars depict arithmetic means. DESeq2 calculated adjusted p-values are shown when < 0.05 for the indicated pairwise comparisons. (**C**) Intracellular c-MYC or control isotype staining of FrB pro-B cells in different stages of the cell cycle identified as shown in *Figure 6—figure supplement 1A*. (**D**) Median fluorescence intensities (MFI) from the staining shown in C. Each point shows data from an individual mouse. Bars show means. P-values were calculated by two-way ANOVA with Tukey's multiple comparisons test. Summary adjusted p value *<0.05, **<0.01, ***<0.001, ****<0.0001. Data shown are from one representative out of two independent experiments. (**E**)

*Figure 8 continued on next page*

*Figure 8 continued*

PTBP1 iCLIP and mRNAseq data visualisation. For PTBP1 iCLIP, x-link sites are shown for all events. Clusters of PTBP1 binding are shown when found. mRNAseq data from pro-B cells of one replicate per genotype is shown using sashimi plot visualisation from Integrative Genomics Viewer (IGV). Numbers on the left show the maximum number of reads in the coverage plots. Arcs depict exon-exon junctions detected in mRNAseq reads. Numbers in arcs show the number of reads for the depicted exon-exon junction. Parts of transcript isoforms predicted to be degraded by NMD are shown in yellow. Parts of transcript isoforms coding for proteins are shown in blue. Exon numbers of transcript isoforms coding for proteins are shown with E and a number. (F) mRNAseq visualisation as in (E) from HeLa control cells or cells where PTBP1 and PTBP2 were knocked down (*Ling et al., 2016*). (G) Human *RBL1* normalised DESeq2 read counts from the two mRNAseq libraries (one control and one double knock down) shown in (F). The online version of this article includes the following source data and figure supplement(s) for figure 8:

**Source data 1.** DESeq2 results for genes shown to have high mRNA expression levels in S or G2/M phases (*Giotti et al., 2019*) in the three pair-wise comparisons shown in *Figure 8A*.
**Figure supplement 1.** PTBP1 binding to target transcripts.
**Figure supplement 2.** c-MYC staining in FrC pro-B cells.

---

cells (*Figure 9A*). CDC25B is a phosphatase promoting G2/M transition by activating CDK1 (*Boutros et al., 2007*). *Cdc25b* AS was unaffected by PTBP1 and PTBP2 absence but PTBP1 bound to the *Cdc25b* 3'UTR (*Figure 4—source data 2* and *Figure 8—figure supplement 1*). Thus, PTBP1 probably promotes *Cdc25b* expression by binding to its 3'UTR and enhancing its stability in pro-B cells.

There were three genes (*Ect2*, *Kif2c* and *Kif22*) whose changes in mRNA abundance could be explained by AS events leading to NMD. ECT2 controls spindle formation in mitosis by exchanging GDP for GTP in small GTPases such as RhoA (*Yüce et al., 2005*). In P1P2dKO pro-B cells an *Ect2* alternative exon with two alternative 3'SS was found more frequently included compared to P1sKO and control pro-B cells (*Figure 9B*). Inclusion of this AS exon, using either of the two alternative 3'SS, will generate transcripts predicted to be degraded by NMD. PTBP1 bound nearby this NMD-triggering exon (*Figure 9B*) where it is predicted to suppress exon inclusion (*Llorian et al., 2010*) promoting high *Ect2* mRNA levels in pro-B cells.

KIF2c (MCAK) and KIF22 (KID) are two kinesin motor family members that transport cargo along microtubules during mitosis (*Cross and McAinsh, 2014*). Both have increased inclusion of an exon that generates predicted NMD-targets in P1P2dKO compared to P1sKO and control pro-B cells and some evidence for PTBP1 binding in adjacent regions (*Figure 9B*). These data indicated that PTBP1 promoted expression of genes important for mitosis by suppressing AS linked to NMD (*Ect2*, *Kif2c*, *Kif22*) and by stabilizing mRNA (*Cdc25b*). The altered expression of these genes in the absence of PTBP1 and PTBP2 is likely to contribute to the high proportions of G2 cells observed in P1P2dKO pro-B cells.

Taken together, our findings implicate PTBP1 as a regulator of the cell cycle in pro-B cells. PTBP1 is essential to control appropriate expression of an mRNA regulon dictating CDK activity, progression to S-phase and entry into mitosis (*Figure 9C*). In the absence of PTBP1 and its partially redundant paralog PTBP2, the molecular control of the cell cycle in primary pro-B cells is disrupted and B cell development is halted.

## Discussion

Here we present an essential role for PTBP1 in controlling B cell development in the bone marrow that is compensated for by PTBP2, but not by PTBP3. Combined PTBP1 and PTBP2 deficiency resulted in a complete block in B cell development and striking defects at two stages of the cell cycle of pro-B cells. The enhanced entry into S-phase of pro-B cells was only observed when both PTBP1 and PTBP2 were absent. The role of PTBP1 in suppressing S-phase entry was unanticipated since in other systems, including GC B cells, PTBP1 promoted proliferation (*Suckale et al., 2011*; *Shibayama et al., 2009*; *La Porta et al., 2016*) and progression through late S-phase (*Monzón-Casanova et al., 2018*). However, these previous studies were done in the presence of PTBP2 (*Suckale et al., 2011*; *Shibayama et al., 2009*; *La Porta et al., 2016*) and GC B cells did not tolerate deletion of both PTBP1 and PTBP2 (*Monzón-Casanova et al., 2018*). Thus, the role of PTBP1 in supressing entry into S-phase may not be unique to pro-B cells and may be found in other cell types if they survive the absence of PTBP1 and PTBP2 long enough to assess cell-cycle progression. Similar

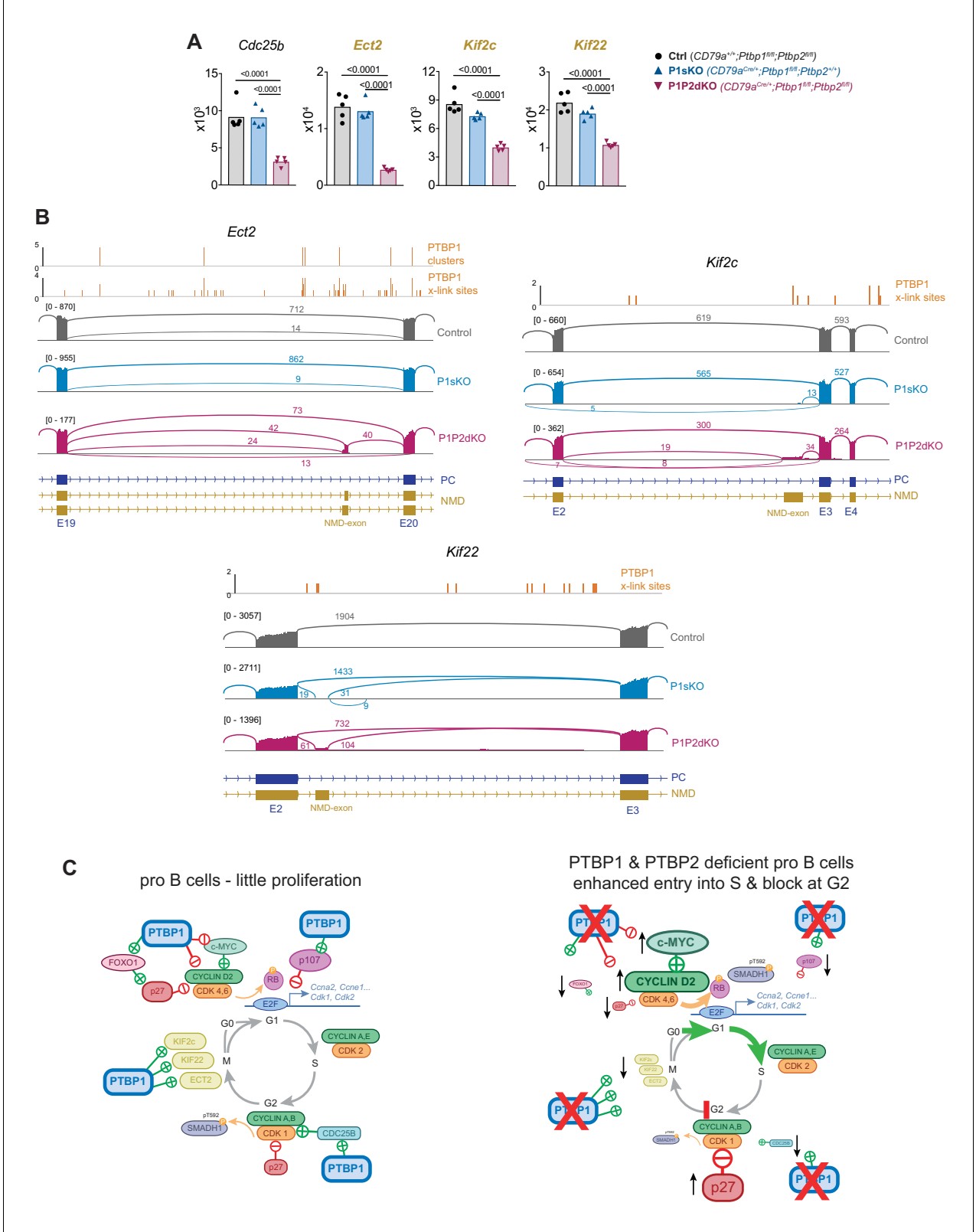

**Figure 9.** PTBP1 controls expression of genes important for mitosis. (**A**) mRNA abundance of the indicated genes. Genes whose names are in yellow and bold are predicted to have reduced mRNA abundance due to changes in AS triggering NMD upon *Ptbp1* and *Ptbp2* deletion. Individual data points show DESeq2 normalised read counts from individual pro-B cell mRNAseq libraries. Bars depict arithmetic means. DESeq2 calculated adjusted p-values are shown when < 0.05 for the indicated pairwise comparisons. (**B**) PTBP1 iCLIP and mRNA-Seq visualisation as described in *Figure 8E*. *Ect2*

*Figure 9 continued*

has an additional isoform that appears with low numbers of reads which is not depicted in the gene model. This isoform results from the inclusion of an alternative 3' splice site on *Ect2* exon 20 which will generate in an isoform predicted to be degraded by NMD. (**C**) Representation of cell cycle mRNA regulon controlled by PTBP1 in pro-B cells and consequences of *Ptbp1* and *Ptbp2* deletion in pro-B cells. Depicted interactions of PTBP1 with individual factors are likely to be direct.

to our findings in pro-B cells, PTBP1-deficient GC B cells and embryonic stem cells also had increased proportions of cells in G2/M phases of the cell cycle (*Monzón-Casanova et al., 2018*; *Shibayama et al., 2009*). Therefore, the role of PTBP1 in promoting G2 transition into mitosis is conserved in other systems.

Although the absence of PTBP1 and PTBP2 caused qualitative and quantitative changes in mRNA expression of many genes, we found that PTBP1 controlled the abundance and splicing of a collection of mRNAs that together comprise a cell cycle mRNA regulon. We identified direct binding of PTBP1 to the targets of this cell cycle mRNA regulon using mature proliferating B cells. Shared and specific mechanisms will control the cell cycle in mature and pro-B cells but the components of this cell cycle mRNA regulon are expressed in both mature and pro-B cells (*Table 1*). Therefore, direct binding of PTBP1 to these targets is expected in pro-B cells. The proteins encoded within this mRNA regulon interact with each other to generate feed-forward activation loops that promote cell cycle entry and mitosis. For example, c-MYC suppresses *Cdkn1b* (encoding p27) (*Yang et al., 2001*) and increases *Ccnd2* transcription (*Bouchard et al., 2001*). CYCLIN-D2 promotes nuclear export and degradation of p27/CDKN1b (*Susaki et al., 2007*) and p27 directly inhibits CYCLIN-D-CDK4/6 complexes (*Yoon et al., 2012*). Although the block in G2-phase we observed could have resulted from a response to DNA damage we found little evidence for enhanced DNA damage in FrB pro-B cells (data not shown). This data suggested that the block at G2 in P1P2dKO pro-B cells resulted mainly from increased p27 expression. The net outcome from deregulation of the PTBP1 controlled cell cycle mRNA regulon in the absence of PTBP1 and PTBP2 is an enhanced CDK activity in G0/G1 pro-B cells that drives entry into S-phase and a reduced CDK activity amongst blocked G2 pro-B cells.

Our study focused on FrB pro-B cells to increase the likelihood of identifying direct targets of PTBP1 as this is the first stage in B cell development where *Ptbp1* and *Ptbp2* were deleted with the *CD79a^cre^* transgene. Absence of PTBP1 and PTBP2 in FrC pro-B cells resulted in the same defects in proliferation as in FrB pro-B cells. Thus, alterations in the progression through the cell cycle in both fractions contributed to the block in B cell development in the whole pro-B cell population. FrC pro-B cells lacking PTBP1 and PTBP2 also had an increase in c-MYC expression at the G0/G1 stage and increased p27 expression at G2. Therefore, the molecular mechanisms by which PTBP1 controls the cell cycle and promotes B cell development in pro-B cells were conserved in both pro-B cell fractions.

**Table 1.** mRNA expression levels of cell cycle regulators in mitogen activated mature B cells and FrB pro-B cells.
Values shown are arithmetic mean Transcripts Per Million (TPMs) from four and five mRNAseq libraries for activated mature B cells and for FrB pro-B cells, respectively.

|  | Activated mature B cells | FrB pro-B cells |
|---|---|---|
| *Myc* | 155.8 | 315.3 |
| *Ccnd2* | 128.5 | 52.7 |
| *Btg2* | 51.5 | 109.3 |
| *Rbl1* | 19.3 | 127.0 |
| *Cdc25b* | 66.8 | 190.1 |
| *Ect2* | 17.8 | 198.9 |
| *Kif2c* | 30.0 | 151.6 |
| *Kif22* | 35.6 | 344.3 |

The molecular mechanisms of PTBP action on the cell cycle mRNA regulon that we assessed were limited to the regulation of AS and mRNA abundance. PTBPs are known, in addition, to control poly-adenylation site choice and IRES-mediated translation of certain transcripts (*Hu et al., 2018*). Therefore, we can expect that additional roles for PTBP1 regulating translation efficiencies and polyadenylation site usage in pro-B cells may emerge. Moreover, by suppressing the inclusion of non-conserved cryptic exons that generate NMD targets (*Ling et al., 2016*; *Sibley et al., 2016*), PTBPs may play a broader role in maintaining the global 'fidelity' of the transcriptome. Cryptic exons, that are poorly conserved between mouse and human have been found when both PTBP1 and PTBP2 were depleted (*Ling et al., 2016*). In pro-B cells deficient for PTBP1 and PTBP2 we found cryptic exons promoting mRNA degradation amongst cell cycle regulators. For most of them we did not find evidence for their conservation in human cells where PTBP1 and PTBP2 were absent (*Ling et al., 2016*). However, the A5SS in *Rbl1* that generates an NMD-target was conserved in human, suggesting that it is a functional AS-NMD event with a role in post-transcriptional control.

The role that we have found here for PTBP1 and PTBP2 is distinct from that of other RBPs such as AUF1 (*Sadri et al., 2010*) and ELAVL1 (*Diaz-Muñoz et al., 2015*) which, when deleted, have little effect on B cell development. The ZFP36 family members ZFP36L1 and ZFP36L2 are important for quiescence during lymphocyte development, but their absence has a much milder impact on B cell development (*Galloway et al., 2016*) than does the combined absence of PTBP1 and PTBP2. The unique role that we have found for PTBP1 in controlling the cell cycle, and its redundancy with PTBP2, further underscores the importance of post-transcriptional RNA regulation as an essential component of the molecular regulation of the cell cycle. In the context of B cells this regulation is essential for developmental progression beyond the pro-B cell stage.

## Materials and methods

### Mice

Mice were bred and maintained in the Babraham Institute Biological Support Unit. Since the opening of this barrier facility (2009), no primary pathogens or additional agents listed in the FELASA recommendations have been confirmed during health monitoring surveys of the stock holding rooms. Ambient temperature was ~19–21°C and relative humidity 52%. Lighting was provided on a 12 hr light: 12 hr dark cycle including 15 min 'dawn' and 'dusk' periods of subdued lighting. After weaning, mice were transferred to individually ventilated cages with 1–5 mice per cage. Mice were fed CRM (P) VP diet (Special Diet Services) ad libitum and received seeds (e.g. sunflower, millet) at the time of cage-cleaning as part of their environmental enrichment. All mouse experimentation was approved by the Babraham Institute Animal Welfare and Ethical Review Body. Animal husbandry and experimentation complied with existing European Union and United Kingdom Home Office legislation and local standards. All mice were used experimentally between 8 and 15 weeks of age and were age- and sex-matched within experiments, although no sex-associated differences were observed in the results obtained. *Ptbp3*$^{fl}$ mice were generated by inserting LoxP sites in either side of *Ptbp3* exon eight to ensure deletion of all PTBP3 isoforms (*Tan et al., 2015*). Embryonic stem cells targeted correctly by homologous recombination were identified by Southern blotting with 5' and 3' probes flanking exon 8. Chimeras were generated using standard techniques. Neomycin cassette deletion was carried out by crossing the mice to C57BL/6 Flp deleter mice. Conditional knockout mice used were derived from crossing the following transgenic strains: *Ptbp1*$^{fl/fl}$ (Ptbp1$^{tm1Msol}$) (*Suckale et al., 2011*), *Ptbp2*$^{fl/fl}$ (Ptbp2$^{tm1.1Dblk}$) (*Li et al., 2014*), *Ptbp3*$^{fl/fl}$ (described here) and *Cd79a*$^{cre}$ (Cd79a$^{tm1(cre)Reth}$) (*Hobeika et al., 2006*). *Rag2*$^{-/-}$ knockout mice (*Shinkai et al., 1992*) (Rag2$^{tm1Fwa}$) were also used. All mice were on the C57BL/6 background.

### In vivo EdU and BrdU administration

In EdU-only and in EdU and BrdU double-labelling experiments 1 to 5 mice of different genotypes and the same sex were kept in individually ventilated cages. Whenever possible females were used as this allowed for a higher number of mice with different genotypes per cage. Males and females showed the same phenotypes observed due to PTBP absence. In EdU-only labelling experiments mice were injected with 1 mg EdU (5-ethynyl-2'-deoxyuridine, cat #E10415, ThermoFisher Scientific) intraperitoneally and were killed one hour after injection. In EdU and BrdU (5-bromo-2'-

deoxyuridine, cat# B5002-500mg, Sigma) labelling experiments mice were injected first with 1 mg EdU (cat #E10415, ThermoFisher Scientific) intraperitoneally. One hour later, the same mice were injected with 2 mg BrdU and the mice were killed 1 hr and 40 min after the injection with EdU.

## Flow cytometry

Single cell suspensions were prepared from spleens and lymph nodes by passing the organs through cell strainers with 70 µm and 40 µm pore sizes in cold RPMI-1640 (cat# R8758, Sigma) with 2% fetal calf serum (FCS). Single cell suspensions from bone marrow were prepared by flushing the marrow from femurs and tibias and passing the cells through a cell strainer with 40 µm pore size in cold RPMI-1640 with 2%FCS. Fc receptors were blocked with monoclonal rat antibody 2.4G2. Cells were stained with combinations of antibodies listed in the Key Resources Table (*Supplementary file 3*). Cell surface staining was carried out for 45 min on ice by incubating cells with a mixture of antibodies in cold FACS buffer (PBS +0.5%FCS). For intracellular staining, cells were fixed with Cytofix/Cytoperm Fixation and Permeabilization Solution (cat# 554722, BD) on ice, washed with FACS Buffer and frozen in 10% DMSO 90% FCS at −80°C at least overnight. After thawing, cells were washed with FACS Buffer and re-fixed with Cytofix/Cytoperm Fixation and Permeabilization Solution (cat# 554722, BD) for 5 min on ice. Cells were washed with Perm/Wash Buffer (cat# 554723, BD) and intracellular staining with antibodies was carried out by incubating fixed and permeabilized cells in Perm/Wash Buffer (cat# 554723, BD) first with monoclonal rat antibody 2.4G2 and subsequently with the desired antibodies in Perm/Wash Buffer at room temperature. EdU was detected with Click-iT Plus EdU kits (cat# C10646, for AF594 and cat# C10636 for pacific blue, ThermoFisher Scientific). For double detection of EdU and BrdU, cells were treated as for intracellular staining but, before adding the intracellular antibodies, cells were treated with TURBO DNase (12 units/107 cells, cat# AM2239, ThermoFisher Scientific) for 1 hr at 37°C. Subsequently, EdU was detected with the Click-iT reaction following the instructions from the manufacturer. Cells were washed with Perm/Wash Buffer (cat# 554723, BD) and incubated with anti-BrdU-AF647 antibody (MoBU-1, cat# B35133, ThermoFisher Scientific). DNA was stained in the last step before flow cytometry analysis with 7AAD in EdU and BrdU double-labelling experiments or with Vybrant DyeCycle Violet Stain in experiments where no DNA-digestion was carried out (cat# V35003, ThermoFisher Scientific). Flow cytometry data were acquired on a BD LSRFortessa with five lasers and was analysed using FlowJo software (versions 10.6.0 and 10.0.8r1).

## mRNAseq libraries from FrB pro-B cells

FrB c-KIT+ pro-B cells (B220+, CD19+, IgD-, IgM-, CD2-, CD25-, CD43high, cKIT+, CD24+ and CD249-) were sorted from bone marrow cells isolated from femurs and tibias. Bone marrow cells from 4 to 6 mice of the same genotype and sex were pooled and depleted of unwanted cells with anti Gr-1 (RB6-8C5), CD11b (M1/70), IgD (11–26 c.2a), NK1.1 (PK136), CD3e (145–2 C11) and Ter119 biotinylated antibodies and anti-biotin microbeads (cat# 130-090-485, Miltenyi) before sorting. The sorting strategy for FrB pro-B cells is shown in *Figure 4—figure supplement 1*. RNA from 15,000 to 200,000 FrB cells was isolated with the RNeasy Micro Kit (cat# 74004, Qiagen). mRNAseq libraries were prepared from five biological replicates per genotype: (three genotypes: ctrl (*Cd79a+/+*; *Ptbp1fl/fl;Ptbp2fl/fl*), P1sKO (*Cd79acre/+;Ptbp1fl/f;lPtbp2+/+*) and P1P2dKO (*Cd79acre/+;Ptbp1fl/fl*; *Ptbp2fl/fl*); three samples from females and two samples from males per genotype) by generating cDNA from 2 ng RNA and 9 PCR cycles per replicate with the SMART-Seq v4 Ultra Low Input RNA Kit for Sequencing (cat# 634891, Takara) and by enzymatic fragmentation of 300 pg of cDNA followed by 12 PCR cycles using the Nextera XT DNA Library Preparation Kit (cat# FC-131–1096, Illumina). The reads that provide the most useful information from short read RNAseq experiments to asses AS are the reads that after mapping to the genome span two separate exons, as these inform of the exon-exon junctions and splice sites used. The probabilities of capturing such exon-exon spanning reads increase with sequencing depth and length. Therefore, we sequenced the mRNAseq libraries with an Illumina HiSeq2500 on a 2 × 125 bp paired-end run at high depth. We obtained an average of 52.5 million reads per sample from each end resulting in an average of ~105 total million reads per sample.

## PTBP1 iCLIP

PTBP1 iCLIP was carried out previously from mitogen-activated B cells (splenic B cells stimulated with LPS for 48 hr) (*Monzón-Casanova et al., 2018*). The data generated were re-analysed to map reads using a splicing-aware software. Reads from five PTBP1 iCLIP libraries were mapped to the GRCm38.p5 mouse genome from Gencode with STAR (v 2.5.4b) (*Dobin et al., 2013*). Reads were de-duplicated using random barcodes included in the library preparation and xlink-sites (*Supplementary file 1*) and clusters of binding (*Supplementary file 2*) were identified with iCount https://icount.readthedocs.io/en/latest/cite.html as previously described (*Monzón-Casanova et al., 2018*). Detection of a binding site with iCLIP (*König et al., 2010*) is highly dependent on the abundance of the RNA, therefore all replicates were pooled together to identify xlink sites and clusters of binding. Xlink sites and clusters of PTBP1 binding were assigned to transcripts and genomic features with the following hierarchy: CDS, 3'UTR, 5'UTR, intron, ncRNA using the Mus_musculus. GRCm38.93.gtf annotation from Ensembl (*Supplementary files 1* and *2*). mRNA abundance analysis mRNAseq libraries were trimmed with Trim Galore (v 1.15 https://www.bioinformatics.babraham.ac. uk/projects/trim_galore/) with default parameters and mapped with Hisat2 (v 2.1.0) (*Kim et al., 2015*) with -p 7 t —phred33-quals —no-mixed —no-discordant parameters, the Mus_musculus. GRCm38 genome and known splice sites from the Mus_musculus.GRCm38.90 annotation. Read counts mapped over genes were counted with HTSeq (v0.10.0) (*Anders et al., 2015*) with f bam -r name -s no -a 10 parameters and the Mus_musculus.GRCm38.93.gtf annotation from Ensembl. Reads mapping to immunoglobulin genes (including V, D, J gene segments, light and heavy immunoglobulin genes and T cell receptor genes) were excluded before DESeq2 analysis. Differences in mRNA abundance were computed with DESeq2 (v1.22.1) (*Love et al., 2014*) by extracting differences between different genotypes in pair-wise comparisons using 'apeglm' method as a shrinkage estimator (*Zhu et al., 2019*). Information on the sex of the mice from which the mRNAseq libraries were generated was included in the design formula in addition to the genotype (design = ~Sex + Genotype). From the DESeq2 results we only considered genes with a mean expression level of at least 1 FPKM (from the five biological replicates) in any of the three genotypes analysed. FPKMs were calculated with cuffnorm from Cufflinks (v2.2.1) (*Trapnell et al., 2012*) using the -library-norm-method geometric. We considered genes with differential mRNA abundance as those with a |log2-fold change| > 0.5 and a p-adjusted value of <0.05 (*Figure 4—source data 1*).

A gene with different mRNA abundance was bound by PTBP1 at the 3'UTR if at least one cluster of PTBP1-binding from the PTBP1 iCLIP data (*Supplementary file 2*) was found to overlap with any 3'UTRs annotated in the Mus_musculus.GRCm38.93.gtf for that gene after assignation to genomic features for the binding sites of the iCLIP as described above.

Genes with different mRNA abundance in P1P2dKO FrB pro-B cells compared to P1sKO and control FrB cells (*Figure 4D* and *Figure 4—source data 1*) were identified after hierarchical clustering of the Euclidian distances between Z-scores of each gene calculated from the DESeq2 normalised read counts for the 15 mRNAseq libraries (five biological replicates per genotype). Genes with different mRNA abundance in any of the three pairwise comparisons carried out (*Figure 4A*) were considered in the hierarchical clustering.

## Alternative splicing analysis

Trimmed reads generated by Trim Galore were further trimmed to 123 bp and reads which were smaller than 123 bp were discarded with Trimmomatic (v0.35) (*Bolger et al., 2014*) in order to obtain only pairs of reads 123 bp long. 123 bp-long reads were mapped to the Mus_musculus. GRCm38 genome as described above, but only uniquely-mapped reads were kept by using the -bS -F 4 F 8 F 256 -q 20 parameters in samtools when converting hisat2 sam files to bam files. rMATS (Turbo v4.0.2) (*Shen et al., 2014*) was run with the -t paired —readLength 123 parameters and the Mus_musculus.GRCm38.93.gtf annotation for each individual pairwise comparison. Only results from rMATS with reads on exon-exon junctions were considered further. Significantly differential alternative splicing events (*Figure 4—source data 2*) were defined as events that have an FDR < 0.05, have an absolute inclusion level difference >0.1 (to reduce the number of significant (FDR < 0.05) differentially spliced events computed by rMATS with small changes in splicing between the different pairwise comparisons), come from genes expressed with at least 1 FPKM (mean across five biological replicates in any of the genotypes analysed) and have at least 80 reads from the sum of the five

biological replicates mapping to either the included or skipped alternative splicing event in at least one of the two conditions analysed. Only alternative splicing events with the highest inclusion level difference were kept out of AS events which had the same genomic coordinates in the AS event (*Figure 4—source data 2*). rMATS considers five different types of alternative splicing events: skipped exons (SE), mutually exclusive exons (MXE), alternative 5' and 3' splice sites (A5SS and A3SS, respectively) and retained introns (RI) (*Figure 4—figure supplement 2C*). Proportions of differential alternative splicing events were defined as bound by PTBP1 (*Figure 4—figure supplement 2D* and *Figure 4—source data 2*) with different criteria depending on the type of AS event.

An SE was bound by PTBP1 if PTBP1 clusters (*Supplementary file 2*) were found on the SE, 500 nucleotides upstream or downstream of the AS SE, on either of the constitutive flanking exons or in 500 nucleotides downstream or upstream of the upstream and downstream constitutive exons, respectively. An A5SS was bound if PTBP1 clusters were found on the longest exon containing the A5SS, on the downstream intronic 500 nucleotides of the A5SS, on the downstream constitutive flanking exon or the 500 intronic nucleotides upstream of the downstream constitutive flanking exon. An A3SS was bound by PTBP1 if PTBP1 clusters were found on the longest exon containing the A3SS, the 500 intronic nucleotides upstream of the A3SS, the upstream constitutive flanking exon or the 500 intronic nucleotides downstream of the upstream constitutive flanking exon. An MXE was bound by PTBP1 if PTBP1 clusters were found on either MXE, the upstream or downstream 500 intronic nucleotides of either MXE, the upstream or downstream constitutive flanking exons or the downstream or upstream 500 intronic nucleotides from the upstream or downstream constitutive flanking exons, respectively. A RI was bound by PTBP1 if PTBP1 clusters were found on the RI or on either constitutive flanking exon.

Events with changes in AS in P1P2dKO FrB pro-B cells compared to control and P1sKO FrB pro-B cells (*Figure 4E*) were identified by hierarchical clustering using complete linkage clustering of the Euclidean distances between Z-scores of each AS event calculated from the rMATS inclusion levels for the 15 mRNAseq libraries (five biological replicates per genotype). AS events found in any of the three pairwise comparisons (*Figure 4B*) were used in the hierarchical clustering.

## Gene ontology term enrichment analysis

Genes belonging to different clusters based on differences in mRNA abundance or AS patterns between the P1P2dKO FrB pro B cells and the other genotypes (P1sKO and Controls) (*Figure 4D and E* and *Figure 5—source data 1*) were used for gene ontology enrichment analysis with GOrilla (*Eden et al., 2009*). Genes expressed with a mean of least 1 FPKM across the five biological replicates in any of the genotypes were used as background list for expressed genes in FrB pro-B cells. *Figure 5D* shows representative enriched terms selected amongst closely related GO terms by manual inspection of the ontology. *Figure 5—source data 1* is a full list of all enriched GO enriched terms. Representative selected terms are highlighted.

## Comparison of transcriptomes from pro-B cells and mitogen activated B cells

Transcriptomes from control FrB pro-B cells and mitogen-activated primary B cells (LPS for 48 hr) (*Diaz-Muñoz et al., 2015*) were compared by calculating Spearman's rank correlation of the Log mean TPM (Transcripts Per Million) for each gene. Mean TPMs were calculated in FrB pro-B cells from five biological replicates and in mitogen-activated B cells from four biological replicates. TPMs were calculated after counting reads mapping to genes with HTSeq and the Mus_musculus. GRCm38.93.gtf annotation from Ensembl.

## mRNAseq from human PTBP1 and PTBP2 depleted HeLa

Sequences from control and mRNAseq libraries where PTBP1 and PTBP2 were knocked down in HeLa cells by *Ling et al. (2016)* were trimmed with Trim Galore (v 0.6.2_dev) and mapped with Hisat2 (v 2.1.0) (*Kim et al., 2015*) with –dta `—sp 1000,1000 p 7 t —phred33-quals —no-mixed — no-discordant` parameters, the Homo_sapiens.GRCh38 genome annotation and a file with known splice sites generated from the Homo_sapiens.GRCh38.87 annotation. Mapped reads were counted with HTSeq (v0.10.0) (*Anders et al., 2015*) with -f bam -r name -s no -a 10 parameters and the

Homo_sapiens.GRCh38.93.gtf annotation from Ensembl. Read counts were normalised with DESeq2 (*Love et al., 2014*) (v 1.20.0).

## Statistical analysis

Statistical analysis of flow cytometry data was carried out with GraphPad Prism version 7.0e. Details of tests carried out are found in the legends. Statistical analysis of mRNAseq data was carried out as described in the 'mRNA abundance analysis' and 'Alternative splicing analysis' sections.

Data availability mRNAseq libraries and iCLIP analysis generated in this study have been deposited in GEO and can be accessed with the GSE136882 accession code at GEO. Mitogen-activated primary B cell mRNAseq libraries were previously reported and can be accessed with the GSM1520115, GSM1520116, GSM1520117and GSM1520118 accession codes in GEO.

## Acknowledgements

We thank Douglas L Black for the *Ptbp2*<sup>fl/fl</sup> mice, Michael Reth for the *Cd79a*<sup>cre</sup> mice, Frederick W Alt for the *Rag2*<sup>-/-</sup> mice and Michele Solimena for the *Ptbp1*<sup>fl/fl</sup> mice and the anti-PTBP2 antibody; Kirsty Bates, Arthur Davis, Attila Bebes, Simon Andrews, the Babraham Institute Biological Support Unit, Flow Cytometry and Bioinformatics Facilities for technical assistance; Tony Ly, Anne Corcoran, Geoff Butcher, Daniel Hodson and members of the Turner, Smith and Zarnack laboratories for helpful discussions.

## Additional information

### Funding

| Funder | Grant reference number | Author |
|---|---|---|
| Biotechnology and Biological Sciences Research Council | BB/J00152X/1 | Martin Turner |
| Biotechnology and Biological Sciences Research Council | BB/P01898X/1 | Elisa Monzón-Casanova Martin Turner |
| Biotechnology and Biological Sciences Research Council | BBS/E/B/000C0407 | Martin Turner |
| Biotechnology and Biological Sciences Research Council | BBS/E/B/000C0427 | Martin Turner |
| Wellcome | 200823/Z/16/Z | Martin Turner |
| European Cooperation in Science and Technology | CA17103 | Elisa Monzón-Casanova |
| German Research Foundation | ZA 881/2-1 | Kathi Zarnack |

The funders had no role in study design, data collection and interpretation, or the decision to submit the work for publication.

### Author contributions

Elisa Monzón-Casanova, Conceptualization, Investigation, Methodology; Louise S Matheson, Kathi Zarnack, Investigation, Methodology; Kristina Tabbada, Methodology; Christopher WJ Smith, Conceptualization, Supervision, Funding acquisition; Martin Turner, Conceptualization, Supervision, Funding acquisition, Project administration

### Author ORCIDs

Elisa Monzón-Casanova (iD) https://orcid.org/0000-0001-6617-6138
Kathi Zarnack (iD) http://orcid.org/0000-0003-3527-3378
Christopher WJ Smith (iD) http://orcid.org/0000-0002-2753-3398
Martin Turner (iD) https://orcid.org/0000-0002-3801-9896

### Ethics

Animal experimentation: Mice were bred and maintained in the Babraham Institute Biological Support Unit. Since the opening of this barrier facility (2009), no primary pathogens or additional agents listed in the FELASA recommendations have been confirmed during health monitoring surveys of the stock holding rooms. Ambient temperature was ~19-21°C and relative humidity 52%. Lighting was provided on a 12-hour light: 12-hour dark cycle including 15 min 'dawn' and 'dusk' periods of subdued lighting. After weaning, mice were transferred to individually ventilated cages with 1-5 mice per cage. Mice were fed CRM (P) VP diet (Special Diet Services) ad libitum and received seeds (e.g. sunflower, millet) at the time of cage-cleaning as part of their environmental enrichment. All mouse experimentation was approved by the Babraham Institute Animal Welfare and Ethical Review Body (UK Home Office Project Licence /P4D4AF812). Animal husbandry and experimentation complied with existing European Union and United Kingdom Home Office legislation and local standards.

### Decision letter and Author response

Decision letter https://doi.org/10.7554/eLife.53557.sa1
Author response https://doi.org/10.7554/eLife.53557.sa2

## Additional files

### Supplementary files

- Supplementary file 1. PTBP1 binding sites (xlinks).
- Supplementary file 2. PTBP1 binding sites (clusters).
- Supplementary file 3. Key resources table.
- Transparent reporting form

### Data availability

mRNAseq libraries and iCLIP analysis generated in this study have been deposited in GEO and can be accessed with the GSE136882 accession code at GEO. Mitogen-activated primary B cell mRNAseq libraries were previously reported and can be accessed with the GSM1520115, GSM1520116, GSM1520117 and GSM1520118 accession codes in GEO.

The following dataset was generated:

| Author(s) | Year | Dataset title | Dataset URL | Database and Identifier |
|---|---|---|---|---|
| Monzón-Casanova E, Matheson LS, Tabbada K, Zarnack K, Smith CJ, Turner M | 2020 | Polypyrimidine tract binding proteins are essential for B cell development | https://www.ncbi.nlm.nih.gov/geo/query/acc.cgi?acc=GSE136882 | NCBI Gene Expression Omnibus, GSE136882 |

The following previously published datasets were used:

| Author(s) | Year | Dataset title | Dataset URL | Database and Identifier |
|---|---|---|---|---|
| Diaz-Muñoz MD, Bell SE, Fairfax K, Monzon-Casanova E, Cunningham AF, Gonzalez-Porta M, Andrews SR, Bunik VI, Zarnack K, Curk T, Kontoyiannis DL, Ule J, Turner M | 2015 | WT_LPS4 | https://www.ncbi.nlm.nih.gov/geo/query/acc.cgi?acc=GSM1520118 | NCBI Gene Expression Omnibus, GSM1520118 |
| Diaz-Muñoz MD, Bell SE, Fairfax K, Monzon-Casanova | 2015 | WT_LPS1 | https://www.ncbi.nlm.nih.gov/geo/query/acc.cgi?acc=GSM1520115 | NCBI Gene Expression Omnibus, GSM1520115 |

| | | | | |
|---|---|---|---|---|
| E, Cunningham AF, Gonzalez-Porta M, Andrews SR, Bunik VI, Zarnack K, Curk T, Kontoyiannis DL, Ule J, Turner M | | | | |
| Diaz-Muñoz MD, Bell SE, Fairfax K, Monzon-Casanova E, Cunningham AF, Gonzalez-Porta M, Andrews SR, Bunik VI, Zarnack K, Curk T, Kontoyiannis DL, Ule J, Turner M | 2015 | WT_LPS2 | https://www.ncbi.nlm.nih.gov/geo/query/acc.cgi?acc=GSM1520116 | NCBI Gene Expression Omnibus, GSM1520116 |
| Diaz-Muñoz MD, Bell SE, Fairfax K, Monzon-Casanova E, Cunningham AF, Gonzalez-Porta M, Andrews SR, Bunik VI, Zarnack K, Curk T, Kontoyiannis DL, Ule J, Turner M | 2015 | WT_LPS3 | https://www.ncbi.nlm.nih.gov/geo/query/acc.cgi?acc=GSM1520117 | NCBI Gene Expression Omnibus, GSM1520117 |
| Diaz-Muñoz MD, Bell SE, Fairfax K, Monzon-Casanova E, Cunningham AF, Gonzalez-Porta M, Andrews SR, Bunik VI, Zarnack K, Curk T, Kontoyiannis DL, Ule J, Turner M | 2015 | HuR- dependent regulation of mRNA splicing is essential for the B cell antibody response | https://www.ncbi.nlm.nih.gov/geo/query/acc.cgi?acc=GSE62129 | NCBI Gene Expression Omnibus, GSE62129 |
| Ling JP, Chhabra R, Merran JD, Schaughency PM, Wheelan SJ, Corden JL, Wong PC | 2016 | PTBP1 and PTBP2 Repress Nonconserved Cryptic Exons | https://www.ncbi.nlm.nih.gov/bioproject/PRJNA309732 | NCBI BioProject, PRJNA309732 |

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
