## [Decision Letter]

**Acceptance summary:**

Monzon-Casanova, Turner and colleagues examine the role the Polypyrimidine Tract Binding proteins 1 and 2 in B cell development. They demonstrate that a PTBP1 dependent posttranscriptional regulatory program controls progression through B cell development. The PTBP1 proteins have primarily been studied in other cell types, and characterizing their role in hematopoiesis and in cell cycle control is new and significant.

**Decision letter after peer review:**

Thank you for submitting your article "Polypyrimidine tract binding proteins are essential for B cell development" for consideration by *eLife*. Your article has been reviewed by three peer reviewers, one of whom is a member of our Board of Reviewing Editors, and the evaluation has been overseen by James Manley as the Senior Editor. The reviewers have opted to remain anonymous.

The reviewers have discussed the reviews with one another and the Reviewing Editor has drafted this decision to help you prepare a revised submission.

Summary:

This study from Monzon-Casanova, Turner and colleagues examines the role the Polypyrimidine Tract Binding proteins 1 and 2 in B cell development. The Turner group previously showed that the RNA-binding protein PTBP1 plays an important role in the selection of B cells with high-affinity BCRs in the light zone of germinal centers. In the present manuscript, they demonstrate that the conditional ablation of PTBP1 in B cells resulted in increased expression of PTBP2, which can compensate for the loss of PTBP1, as had been observed in other systems. Conditional loss of both PTBP1 and PTBP2 in Cd79a-Cre Ptbp1(fl/fl) Ptbp2(fl/fl) mice (cDKO) led to the arrest of early B cell development at the pro-B cell stage. Detailed analysis revealed that the developmental block was likely due to the deregulation of cell cycle progression in pro-B cells. The entry into S-phase was accelerated, while progression past the G2-phase was arrested. RNA-seq and iCLIP experiments identified PTBP1/2 targets. The accelerated S-phase entry was possibly due to an increase in positive regulators (Myc and Ccnd2) and a decrease in negative regulators (Btg2 and Rbl1/p107) of S-phase entry in cDKO pro-B cells. A strong increase in expression of the CDK1 inhibitor p27 at the G2/M phase transition likely explains the block to cell progression past the G2-phase.

The reviewers all found the paper to be of interest for its findings in both B cell biology and posttranscriptional gene regulation. The demonstration that a posttranscriptional regulatory program controls progression through B cell development is novel and important. The PTBP1 proteins have primarily been studied in other cell types, and characterizing their role in hematopoiesis and in cell cycle control is significant. The methods are fairly standard, and the experiments are carefully executed and controlled. However, a number of questions were raised as to how the results were best interpreted. With proper revision, the study could be a valuable contribution to *eLife*.

Essential revisions:

1) It is not clear why RNA-seq was performed on Fraction B pro-B cells, because the block in the double knockout mice appears to be at the Fraction C to C' transition. The observed gene expression changes seem to have minimal impact on the size of the pro-B cell compartment (Figure 1D). This raises the question of how informative it is to segregate the cells based on CD249/BP1 expression. How would the conclusions be affected if they define pro-B cells as Kit+CD19^+^B220+IgM-IgD-CD25- cells, which encompasses both Fr.B and Fr.C cells?

2) Given that the block appears later than the changes in cell cycle gene expression, what other gene changes may be relevant to the observed Fraction C to C' block? For example, are genes affecting cell survival also altered in the double knockout pro-B cells, which might lead to an increase pro-B cell death? Is expression normal in double knockout pro-B cells for other genes known to be involved in the pro-B to pre-B cell transition? The above analysis may have already been performed, with the conclusion that aberrant expression of cell cycle regulators are the only genetic changes observed. If so, the paper should present a hypothesis for how cell cycle defects in Fraction B pro B cells lead to the Fraction C to C' block.

3) The authors use iCLIP data obtained with mature B cells for the interpretation of the pro-B cell expression data, evidently due to the difficulty in obtaining sufficient pro-B cells to perform iCLIP experiments. This strategy is not without risk, particularly for identifying cell cycle targets that could be under very different control regimes in the mature cells. This should be discussed in more detail.

4) Paragraph two of subsection “PTBP2 can compensate for PTBP1 in B cells”: PTBP2 is not normally expressed during B cell development but a key point is that its expression is induced when PTBP1 is deleted. It would be helpful to show these data in the present paper and emphasize the point more clearly. The only support for this critical information is a reference to Supplemental Data in the previous Nature Immunology paper. Similarly, a discussion of PTBP3 seems oddly missing from the paper, since it is expressed in B cells. Is it doing something completely different? Has it been deleted – does a PTBP1/3 double behave like the PTBP1/2 KO? This might indicate that the proteins have a common function but there is a threshold level needed for the total PTBP1/2/3 protein.

5) More detail is needed on how the RNAseq was done. How many reads were generated per condition? How many replicates? How does this affect the reliability of the splicing analysis? Etc.

6) In other developmental pathways and in cell culture PTBP1 expression is associated with proliferation. Do the authors think that the cell cycle effects upon PTBP loss described here will be general? More discussion of this is needed.

---

## [Author Response]

Essential revisions:1) It is not clear why RNA-seq was performed on Fraction B pro-B cells, because the block in the double knockout mice appears to be at the Fraction C to C' transition. The observed gene expression changes seem to have minimal impact on the size of the pro-B cell compartment (Figure 1D). This raises the question of how informative it is to segregate the cells based on CD249/BP1 expression. How would the conclusions be affected if they define pro-B cells as Kit+CD19^+^B220+IgM-IgD-CD25- cells, which encompasses both Fr.B and Fr.C cells?

FrB pro-B cells are the first stage in development were PTBP1 and PTBP2 are absent in our P1P2 double conditional KO mouse model (P1P2dKO). We focused our analysis on this stage as it enhances the likelihood to find direct targets of PTBP1 necessary for the development of B cells and therefore the segregation between FrB and FrC pro-B cells is informative. We have added this explanation in our revised manuscript (Results section “PTBP1 regulates mRNA abundance and AS in pro-B cells”, “We focused our analysis on FrB pro-B cells…”).

We include new data identifying pro-B cells as Kit+CD19^+^B220+IgM-IgD-CD25- cells. Identification of pro-B cells with this strategy resulted in only slightly reduced numbers of pro-B cells in P1P2dKO mice compared to control mice (new Figure 3—figure supplement 1A). Therefore, our conclusions are the same after defining pro-B cells with this combination of cell surface proteins and by segregating them into FrB and FrC pro-B cells. We discuss this new data in the Results section (“Enumeration of pro-B cells identified as c-Kit+ CD19^+^ B220+ IgM- IgD- CD25- cells yielded similar results (Figure 3—figure supplement 1A).”).

2) Given that the block appears later than the changes in cell cycle gene expression, what other gene changes may be relevant to the observed Fraction C to C' block? For example, are genes affecting cell survival also altered in the double knockout pro-B cells, which might lead to an increase pro-B cell death? Is expression normal in double knockout pro-B cells for other genes known to be involved in the pro-B to pre-B cell transition? The above analysis may have already been performed, with the conclusion that aberrant expression of cell cycle regulators are the only genetic changes observed. If so, the paper should present a hypothesis for how cell cycle defects in Fraction B pro B cells lead to the Fraction C to C' block.

We did not find an enrichment for gene-ontology terms related to cell death or apoptosis and mRNA expression of genes important for cell survival such as *Bcl2, BCl^-^XL (Bcl2l2), Bax and Bim (Bcl2l11)* was similar in P1P2dKO pro-B cells compared to control pro-B cells. We have included a sentence specifying this information in the Results section “PTBP1 regulates pathways associated with growth and proliferation” (“Similarly, genes regulating cell survival and apoptosis such as Bcl2, Bcl2l2 (BCl^-^XL), Bax and Bcl2l11 (Bim) had normal mRNA abundance and AS patterns in P1P2dKO pro-B cells (Figure 4—source data 1 and 2))”).

In the Results section “PTBP1 regulates pathways associated with growth and proliferation” we discuss the expression of genes known to be important for the transition between pro- and pre-B cells shown in (Figure 4—figure supplement 2E). Most of these genes have normal mRNA expression in P1P2dKO pro-B cells and we elaborate further on those that have changes in P1P2dKO pro-B cells compared to control pro-B cells. Our conclusion is that the essential elements of B cell development at the pro-B cell stage are unaffected by the absence of PTBP1 and PTBP2.

In our revised manuscript we present new data showing that the cell cycle defects observed in P1P2dKO FrB pro-B cells were also found in P1P2dKO FrC pro-B cells (new data for FrC pro-B cells: Figure 6—figure supplement 1). Similarly, we show that the expression patterns of c-MYC, p27, pS807/811-RB, pT592-SAMHD1 in FrC P1P2dKO pro-B cells is the same as in FrB P1P2dKO pro-B cells (new data for FrC pro-B cells: Figure 7—figure supplement 1 and Figure 8—figure supplement 2). Therefore, our hypothesis is that the cell cycle defects observed in FrB pro-cells are conserved at the FrC pro-B cell stage where they will also contribute to the block in B cell development seen in the absence of PTBP1 and PTBP2. We have modified the text in the manuscript (“We found an increase in the proportions of FrB and FrC pro-B cells that are in S-phase …”; “These expression patterns of p27 in P1P2dKO FrB pro-B cells were conserved in P1P2dKO FrC pro-B cells (Figure 7—figure supplement 1).” and “Proportions of both p-RB and p-SAMHD1-positive cells were increased in FrB and FrC P1P2dKO G0/G1 pro-B cells compared to control and P1sKO G0/G1 pro-B cells (Figure 7C, 7D, and Figure 7—figure supplement 1B)”; “FrC P1P2dKO pro B cells in G0/G1 also showed increased cMYC-staining compared to control FrC pro-B cells (Figure 8—figure supplement 2)” and included a paragraph in the Discussion section to reflect this hypothesis (“Our study focused on FrB pro-B…”).

We can expect other genes unrelated to cell cycle control with altered expression in P1P2dKO pro-B cells to have a role in B cell development as there were many genes with altered mRNA expression in the absence of PTBP1 and PTBP2 (Figure 4). Nonetheless, the control of progression through the cell cycle in pro-B cells constitutes a major function of PTBP1 in pro-B cells and the dramatic block at G2 in P1P2dKO pro-B cells will contribute to the block in B cell development.

3) The authors use iCLIP data obtained with mature B cells for the interpretation of the pro-B cell expression data, evidently due to the difficulty in obtaining sufficient pro-B cells to perform iCLIP experiments. This strategy is not without risk, particularly for identifying cell cycle targets that could be under very different control regimes in the mature cells. This should be discussed in more detail.

The PTBP1 targets we identified that form a cell cycle mRNA regulon in pro-B cells were expressed in mature proliferating B cells (new Table 1). Therefore, we can expect that PTBP1 will also bind to these targets in pro-B cells, even if PTBP1-binding in mature B cells is not direct proof of binding in pro-B cells. We have discussed this point in more detail as follows:

In the Discussion section, “…We identified direct binding of PTBP1 to the targets of this cell cycle mRNA regulon using mature proliferating B cells. Shared and specific mechanisms will control the cell cycle in mature and pro-B cells. The components of this cell cycle mRNA regulon are expressed in both mature and pro-B cells (Table 1). Therefore, direct binding of PTBP1 to these targets is expected in pro-B cells.”

And in the Results section “PTBP1 regulates pathways associated with growth and proliferation”: “None of these CDK inhibitors has an obvious change in AS or a binding site for PTBP1 in the 3’UTR (Figure 4—source data 1 and 2) although they were expressed in mature B cells (data not shown).”

Moreover, there could be additional PTBP1 targets in pro-B cells that are not captured in our iCLIP from mature proliferating B cells. We also discuss this point further in the Results section “PTBP1 regulates mRNA abundance and AS in pro-B cells”: “The PTBP1 iCLIP data were from mitogen-activated mouse primary B cells because it was not feasible to purify sufficient numbers of pro-B cells for iCLIP. There is a positive correlation between the transcriptomes of mitogen-activated primary B cells and pro-B cells (Figure 4—figure supplement 2A), suggesting that the PTBP1 iCLIP data set (Supplementary Files 1 and 2) is suitable to infer PTBP1-bound RNAs in pro-B cells. We only found 681 genes with no mRNA expression (0 Transcript Per Million (TPM) in activated primary B cells that were expressed in pro-B cells (>= 1 TPM). Therefore, the probability of missing specific targets of PTBP1 in pro-B cells using the iCLIP data from mature activated B cells was small.”

In summary, we acknowledge that this strategy has risks and have discussed them in more detail.

4) Paragraph two of subsection “PTBP2 can compensate for PTBP1 in B cells”: PTBP2 is not normally expressed during B cell development but a key point is that its expression is induced when PTBP1 is deleted. It would be helpful to show these data in the present paper and emphasize the point more clearly. The only support for this critical information is a reference to Supplemental Data in the previous Nature Immunology paper. Similarly, a discussion of PTBP3 seems oddly missing from the paper, since it is expressed in B cells. Is it doing something completely different? Has it been deleted – does a PTBP1/3 double behave like the PTBP1/2 KO? This might indicate that the proteins have a common function but there is a threshold level needed for the total PTBP1/2/3 protein.

To emphasise that PTBP2 is only expressed in B cells in the absence of PTBP1 we now present the data on PTBPs expression throughout B cell development in Figure 1 instead of in the supplementary/expanded data. We also present new flow cytometry data showing that PTBP2 is only expressed in B cells in the absence of PTBP1 in Figure 2B and emphasize this point more clearly in the text (“Deletion of Ptbp1 resulted in expression of PTBP2 (Figure 2B and (Monzon-Casanova et al., 2018))…”). Moreover, we also show now the expression of PTBP2 in PTBP1-deficient pro-B cells in Figure 3C, while in our previous version of the manuscript this data were in a supplementary figure.

We addressed the roles of PTBP3 in B cell development with a novel *Ptbp3*-floxed allele and include our results now in the revised manuscript (“To study the roles of PTBP3 in B cell development we deleted Ptbp3…”). Deletion of *Ptbp3* results in normal B cell development (new data in Figure 2—figure supplement 1). Therefore, PTBP1 compensates for any potential roles of PTBP3 in the development of B cells.

Moreover, we also generated a Ptbp1 and Ptbp3 double conditional KO mouse (P1P3dKO) with the *Ptbp1^fl^, Ptbp3^fl^* and *Cd79a^cre^* alleles. B cell development in the bone marrow of these mice is normal (new data in Figure 3—figure supplement 2A). P1P3dKO B cells express PTBP2 (new data in Figure 3—figure supplement 2B). Therefore, PTBP2 in P1P3dKO B cells compensates for the functions of PTBP1 and PTBP3 during the development of B cells in the bone marrow. These data show high redundancy between the different PTBP paralogs in B cell development in the bone marrow. The different phenotypes observed in the different conditional knockout mouse models could indeed reflect a requirement of certain amounts of total PTBP protein at distinct stages of B cell development. We discuss this possibility in the Results section “The essential role for PTBP1 in the absence of PTBP2 in B cell development is at the pro-B cell stage” in our revised manuscript (“Deletion of Ptbp1 and Ptbp3 at the pro-B cell stage…”)

Importantly, the developmental block at the pro-B cell stage in the P1P2dKO mice despite the expression of PTBP3 by P1P2dKO pro-B cells, shows that PTBP1 is the most important PTBP paralog for successful B cell development in the bone marrow. Therefore, we focus on the roles of PTBP1 in our manuscript.

5) More detail is needed on how the RNAseq was done. How many reads were generated per condition? How many replicates? How does this affect the reliability of the splicing analysis? Etc.

We have included more details on how the RNAseq was carried out in the Results and the Materials and methods section of the manuscript. We carried out five biological replicates per condition.

Materials and methods section “mRNAseq libraries from FrB pro-B cells” new text:

“The reads that provide the most useful information from short read RNAseq experiments to asses AS are the reads that after mapping to the genome span two separate exons, as these inform of the exon-exon junctions and splice sites used. The probabilities of capturing such exon-exon spanning reads increase with sequencing depth and length. Therefore, we sequenced the mRNAseq libraries with an Illumina HiSeq2500 on a 2x125 bp paired-end run at high depth. We obtained an average of 52.5 million reads per sample from each end resulting in an average of ~105 total million reads per sample.”

New text in the Results section “PTBP1 regulates mRNA abundance and AS in pro-B cells”: “We carried out five biological replicates per condition and sequenced the mRNAseq libraries on a 125bp paired-end mode obtaining ~105 million reads per sample to increase the probability of capturing reads spanning two different exons which inform on which splice sites are used”.

6) In other developmental pathways and in cell culture PTBP1 expression is associated with proliferation. Do the authors think that the cell cycle effects upon PTBP loss described here will be general? More discussion of this is needed.

We have extended our discussion related to the roles of PTBP1 in controlling proliferation in the Discussion section. We comment now on the roles of PTBP1 in supressing entry into S-phase and promoting progression from G2 into mitosis separately. “Combined PTBP1 and PTBP2 deficiency resulted in a complete block in B cell development and striking defects at two stages of the cell cycle of pro-B cells. The enhanced entry into S-phase of pro-B cells was only observed when both PTBP1 and PTBP2 were absent. The role of PTBP1 in suppressing S-phase entry was unanticipated since in other systems, including GC B cells, PTBP1 promoted proliferation (Suckale et al., 2011; Shibayama et al., 2009; La Porta et al., 2016) and progression through late S-phase (Monzon-Casanova et al., 2018). However, these previous studies were done in the presence of PTBP2 (Suckale et al., 2011; Shibayama et al., 2009; La Porta et al., 2016) and GC B cells did not tolerate deletion of both PTBP1 and PTBP2 (Monzon-Casanova et al., 2018). Thus, the role of PTBP1 in supressing entry into S-phase may not be unique to pro-B cells and may be found in other cell types if they survive the absence of PTBP1 and PTBP2 long enough to assess cell-cycle progression. Similar to our findings in pro-B cells, PTBP1-deficient GC B cells and embryonic stem cells also had increased proportions of cells in G2/M phases of the cell cycle (Monzon-Casanova et al., 2018; Shibayama et al., 2009). Therefore, the role of PTBP1 in promoting G2 transition into mitosis is conserved in other systems”.